# Co-delivery of IOX1 and doxorubicin for antibody-independent cancer chemo-immunotherapy

Jing Liu[1,2,4], Zhihao Zhao[1,2,4], Nasha Qiu[1], Quan Zhou[1], Guowei Wang[1], Haiping Jiang[3], Ying Piao[1,2], Zhuxian Zhou [1,2], Jianbin Tang[1] & Youqing Shen [1,2 ✉]

Anti-programmed cell death-1 (PD-1)/programmed cell death-ligand 1 (PD-L1) antibodies are currently used in the clinic to interupt the PD-1/PD-L1 immune checkpoint, which reverses T cell dysfunction/exhaustion and shows success in treating cancer. Here, we report a histone demethylase inhibitor, 5-carboxy-8-hydroxyquinoline (IOX1), which inhibits tumour histone demethylase Jumonji domain-containing 1A (JMJD1A) and thus downregulates its downstream β-catenin and subsequent PD-L1, providing an antibody-independent paradigm interrupting the PD-1/PD-L1 checkpoint. Synergistically, IOX1 inhibits cancer cells' P-glycoproteins (P-gp) through the JMJD1A/β-catenin/P-gp pathway and greatly enhances doxorubicin (DOX)-induced immune-stimulatory immunogenic cell death. As a result, the IOX1 and DOX combination greatly promotes T cell infiltration and activity and significantly reduces tumour immunosuppressive factors. Their liposomal combination reduces the growth of various murine tumours, including subcutaneous, orthotopic, and lung metastasis tumours, and offers a long-term immunological memory function against tumour rechallenging. This work provides a small molecule-based potent cancer chemo-immunotherapy.

---

[1] Zhejiang Key Laboratory of Smart Biomaterials and Key Laboratory of Biomass Chemical Engineering of Ministry of Education, College of Chemical and Biological Engineering, Zhejiang University, Hangzhou, China. [2] Hangzhou Global Scientific and Technological Innovation Center, Hangzhou, China. [3] Department of Medical Oncology, The First Affiliated Hospital, School of Medicine, Zhejiang University, Hangzhou, China. [4]These authors contributed equally: Jing Liu, Zhihao Zhao. ✉email: shenyq@zju.edu.cn

mmune checkpoint blockade (ICB) targeting programmed death-1 (PD-1)/programmed death-ligand 1 (PD-L1) axis has dramatically improved the outcomes of cancers[1–3]. In clinics, anti-PD-1/PD-L1 antibodies significantly prolong the patient survival of many cancers, including melanoma[4] and non-small cell lung cancer[5]. The underlying mechanism is that tumour cell's PD-L1 binding the PD-1 on cytotoxic T cells causes their dysfunction and forcefully curbs the immune attack; thus, blocking the PD-1/PD-L1 interaction using antibodies normalises the T cells[6,7]. However, the poor tumour penetration of the large-size antibodies[8] and the tumour immunosuppressive microenvironment cause the ICB therapy ineffective in ~40% melanoma and solid tumours[9–11], particularly large tumours due to further restricted penetration and enhanced immunosuppression[12].

Therefore, relieving the immunosuppressive microenvironment has been explored to enhance the prognosis of anti-PD-1/PD-L1 antibody-therapy[10]. For instance, some chemotherapeutics such as doxorubicin (DOX) can induce tumour cells to undergo immune-stimulatory immunogenic cell death (ICD)[13–16]. The dying cells release danger-associated molecular patterns (DAMPs) and promote the transfer of tumour-associated antigens to dendritic cells (DCs), and thus activate DC maturation and infiltration of T cells and memory T cells to the tumour[17], turning immunosuppressive "cold" tumours to immune-responsive "hot" tumours[10]. Clinical trials at various stages show the benefits of chemotherapy to the ICB-immunotherapy[18–20]. In this combination therapy, low or standard-dose chemotherapy is mostly used to avoid chemotherapy-induced bone marrow suppression and damages to the hosts' immune system[21]. How to efficiently induce ICD at low doses, particularly in drug-resistant cancer cells, is critical to realise effective chemo-immunotherapy[22].

Another key issue to improving the efficacy of the ICB-immunotherapy is to increase the PD-1/PD-L1 blockage efficiency, which is generally restricted due to antibodies' poor tumour penetration. A prospective strategy is to downregulate PD-L1 expression using small molecules. Some kinase inhibitors[23], particularly cyclin-dependent kinase 5 inhibitors[24], and some natural compounds as baicalin[25] are shown to attenuate tumour PD-L1 expression, but their therapeutic efficacies are generally low even in animal models[26,27].

Here, we report a highly potent immunotherapeutic small molecule, 5-carboxy-8-hydroxyquinoline (IOX1), that combined with DOX can reduce the growth of various tumour models (Fig. 1). IOX1 effectively downregulates tumour PD-L1 expression in vitro and in vivo. Simultaneously, it inhibits cancer cell's multidrug resistance and promotes intracellular DOX accumulation, greatly enhancing DOX-induced ICD. Accordingly, the combination of IOX1 with DOX greatly promotes T cell infiltration and activity and significantly reduces tumour immunosuppressive factors. Thus, the liposomal IOX1 and DOX combination eliminates various murine cancer models and produces a long-term immunological memory against subcutaneous (s.c.) and lung metastasis tumour rechallenging. The mechanism study finds that IOX1 inhibits the histone demethylase Jumonji domain-containing 1A, JMJD1A (also called KDM3A or JHDM2A), and thus downregulates β-catenin, thereby reducing its downstream P-gp and PD-L1 expression. Because JMJD1A is overexpressed in many kinds of tumours[28] and promotes cancer cell progression and invasion by epigenetically activating transcription of β-catenin and its target genes[29], the readily available IOX1 provides a potent antibody-free chemo-immunotherapy paradigm for cancer treatment.

## Results

**IOX1 liposome formulation and characterisations**. IOX1 has low cytotoxicity, as confirmed by the MTT assay (Supplementary Fig. 1). At the tested doses, IOX1 alone showed no tumour inhibition activity (Fig. 2a). However, its combination with DOX at a molar ratio of 5: 1 (5IOX1 + DOX) effectively inhibited the tumour growth (Fig. 2a), even significantly better than DOX combination with anti-PD-L1 antibody (αPD-L1). However, the three injections of 5IOX1 + DOX combination did not eradicate the tumours, and the dormant tumours regressed and overgrew once treatment withdrawal. The analysis showed that the intravenously (i.v.) injected IOX1 had a half-life time in blood as short as about 30 min. Thus, slowing down the blood clearance may further enhance the IOX1 adjuvant effect.

We loaded IOX1 into liposome with a similar formula of commercially available pegylated liposomal DOX (PLD, LIBOD®)[30] consisting of DOPE, cholesterol, DSPE-mPEG2000. The prepared IOX1 liposome, noted as IOXL, contained 20.1 wt% of IOX1 and had a very uniform size of 102.3 ± 0.7 nm (Fig. 2b), the same as that of PLD (101.2 ± 0.5 nm; Supplementary Fig. 2). Thus, IOXL was mixed with PLD and obtained a clear solution. The mixed liposome solution of IOXL with PLD at an IOX1 to DOX molar ratio of 5: 1, noted as IPLD, had a small and uniform size of 94.8 ± 0.3 nm and narrow polydispersity (Fig. 2b; Supplementary Fig. 3). IPLD simplifies the i.v. injection for the combination treatment. Both IOXL and IPLD showed excellent long-term stability in various dispersion media (Supplementary Figs. 4–6).

**IOX1 inhibits the multidrug resistance of cancer cells**. IOX1 showed a concentration-dependent inhibitory effect on the P-glycoprotein (P-gp) expression in CT26 and HCT116 cells (Fig. 2c; Supplementary Fig. 7). Moreover, IOX1 also inhibited P-gp ATPase activity (Supplementary Fig. 8), indicating that IOX1 simultaneously downregulated the expression and inhibited the function of P-gp in cancer cells. Therefore, IOX1 treatment significantly enhanced intracellular DOX concentration. As shown in Fig. 2d,e, CT26 cells cultured with free DOX had very little DOX fluorescence even after 8 h, but the presence of IOX1 substantially increased the intracellular DOX level. IOXL had a similar effect on PLD (Supplementary Fig. 9). Flow cytometry analysis revealed that IOX1 or IOXL increased the intracellular DOX fluorescence intensity by 2–3 folds of that of the DOX or PLD group (Fig. 2d,e; Supplementary Fig. 9). Similarly, IOX1 or IOXL could also increase the intracellular DOX fluorescence intensity in HCT116 cells (Supplementary Fig. 10). Another known substrate of P-gp, rhodamine 123, gave similar results (Supplementary Fig. 11). On the contrary, IOX1 could not enhance the DOX accumulation in low P-gp expression NIH-3T3 cells (Supplementary Fig. 12), P-gp-knockdown (Supplementary Figs. 13,14) or -overexpressing CT26 cells (Supplementary Figs. 13,15). These results indicate that IOX1 enhanced the intracellular DOX concentration mainly through downregulating the P-gp expression.

Accordingly, IOX1 or IOXL greatly enhanced the cytotoxicity of DOX against murine cancer cell lines CT26, 4T1, B16F10, and human cancer cell lines HCT116 and MCF-7 cancer cells (Fig. 2f; Supplementary Fig. 16). IOX1 or IOXL alone had low cytotoxicity (Supplementary Fig. 1), but the presence of five times of IOX1 relative to DOX (5IOX1 + DOX) sharply reduced the IC$_{50}$ values of DOX against CT26 cells by 14 times to 19 ng ml$^{-1}$ and that of PLD by ten times to 104 ng ml$^{-1}$. Similarly, the combination with IOX1 could siginificantly decreased the IC$_{50}$ values of DOX against 4T1, B16F10, HCT116 and MCF-7 cells (Fig. 2f; Supplementary Fig. 16). On the contrary, IOX1 had no toxicity to normal cell lines NIH-3T3 and HUVEC cells, and also did not increase the DOX's cytotoxicity to these cells (Supplementary Fig. 17). Moreover, IOX1 also enhanced the cytotoxicity of other

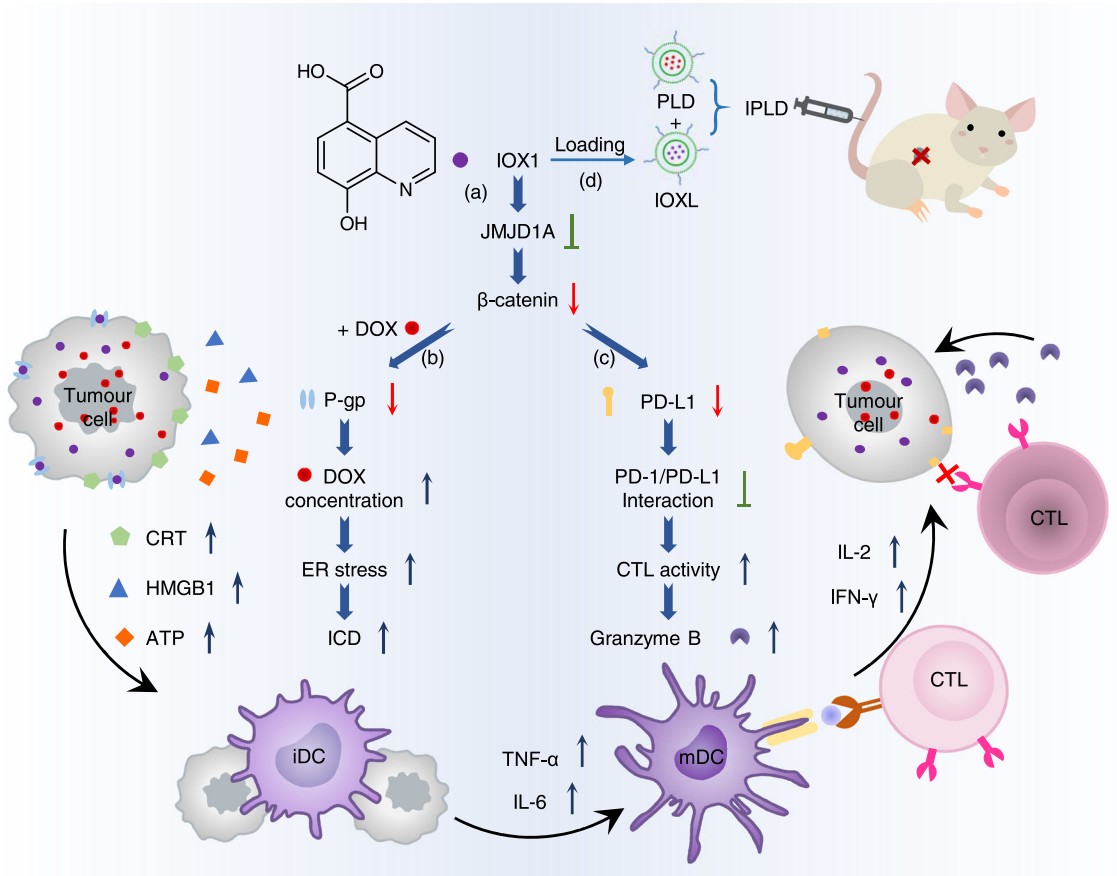

**Fig. 1 5-Carboxy-8-hydroxyquinoline (IOX1) potentiates chemo-immunotherapy. a** IOX1 inhibits the histone demethylase Jumonji domain-containing 1A (JMJD1A) and thus downregulates its downstream β-catenin expression; **b** it downregulates P-glycoproteins (P-gp) and increases doxorubicin (DOX) concentration of cancer cells, facilitating the immunogenic cell death (ICD); **c** it downregulates tumour cells' PD-L1 expression and thus disrupts the PD-1/ PD-L1 pathway with T cells. **d** The synergy of the two effects enables its liposome formula (IOXL) combined with pegylated liposomal DOX (PLD), i.e. IPLD, to inhibit tumour growth. ER: endoplasmic reticulum, CRT: calreticulin, HMGB1: high mobility group box 1, ATP: adenosine triphosphate, iDC: immature dendritic cell, mDC: mature dendritic cell, CTL: cytotoxic T lymphocyte.

chemotherapeutic drugs against tumour cells, such as daunorubicin (DNR), epirubicin (EPI), and oxaliplatin (OXA) (Supplementary Fig. 18).

**IOX1 potentiates DOX to induce ICD**. Cancer cells undergoing ICD present signature molecules, including calreticulin (CRT), high mobility group box 1 (HMGB1), and adenosine triphosphate (ATP)[31–33]. DOX is known to induce ICD of cancer cells[14,34], as confirmed by the increased CRT translocation, reduced nuclear HMGB1, and enhanced autophagy, as well as ATP release, compared with control (Fig. 2g–j; Supplementary Figs. 19–25). IOX1 and IOXL had no ICD-inducing ability; however 5IOX1 + DOX tripled the amounts of membrane-associated CRT (Fig. 2g, h; Supplementary Fig. 19), increased the LC3 granularity and associated ATP release (Fig. 2i; Supplementary Fig. 20), and completely dimmed the HMGB1 staining (Fig. 2g; Supplementary Fig. 21) in CT26 cells due to efficient release from the nuclei to the extracellular fluid. The IPLD had the same, but slightly weaker, effects as the 5IOX1 + DOX combination. CRT membrane translocation and ICD induction are related to the endoplasmic reticulum (ER) stress coping mechanism[35]. Western blotting analysis showed that 5IOX1 + DOX, as well as IPLD, increased the expression of p-PERK and p-eIF2α, i.e. activation of the PERK/eIF2α pathway (Fig. 2j; Supplementary Fig. 22) associated to stronger ER stress, and also promoted the CRT membrane translocation (Fig. 2g,h,j; Supplementary Figs. 19,22). The

same ICD induction was also found in human HCT116 cells (Supplementary Figs. 23–25). These results suggest that IOX1 and its liposome markedly enhanced the DOX's ability to induce immunogenic apoptosis of tumour cells.

**IOX1 mediates the DC maturation**. The released danger molecules are presented to the immune system for recognition and processing by antigen-presenting cells (APCs) such as DCs[36,37]. Thus, the ICD-induced DC maturation was analyzed in vitro using bone marrow-derived dendritic cells (BMDCs) separated from BALB/c mice. CT26 cells were pre-treated with DOX or its combination and then co-incubated with BMDCs; the mature BMDCs (CD11c+CD80+CD86+, mDCs) were counted using flow cytometry (Fig. 3a,b). The population of mDCs in the BMDCs incubated with IOX1-treated CT26 cells was not different from that of the control (BMDCs incubated with non-treated cells). The DOX-treated cells increased mDCs' population to about 29.9%, but the 5IOX1 + DOX-treated cells doubled the mDCs' population to 58.0%. The liposome formula again had the same trend. The maturity of these mDCs was further confirmed by their morphology (Supplementary Fig. 26) and the increased secretion of TNF-α and IL-6 (Fig. 3c). As IOX1 alone showed no effect on DC maturation or secretion of TNF-α and IL-6 (Supplementary Fig. 27), and CRT-knockdown CT26 cells pre-treated with 5IOX1 + DOX could not induce the maturation of BMDCs

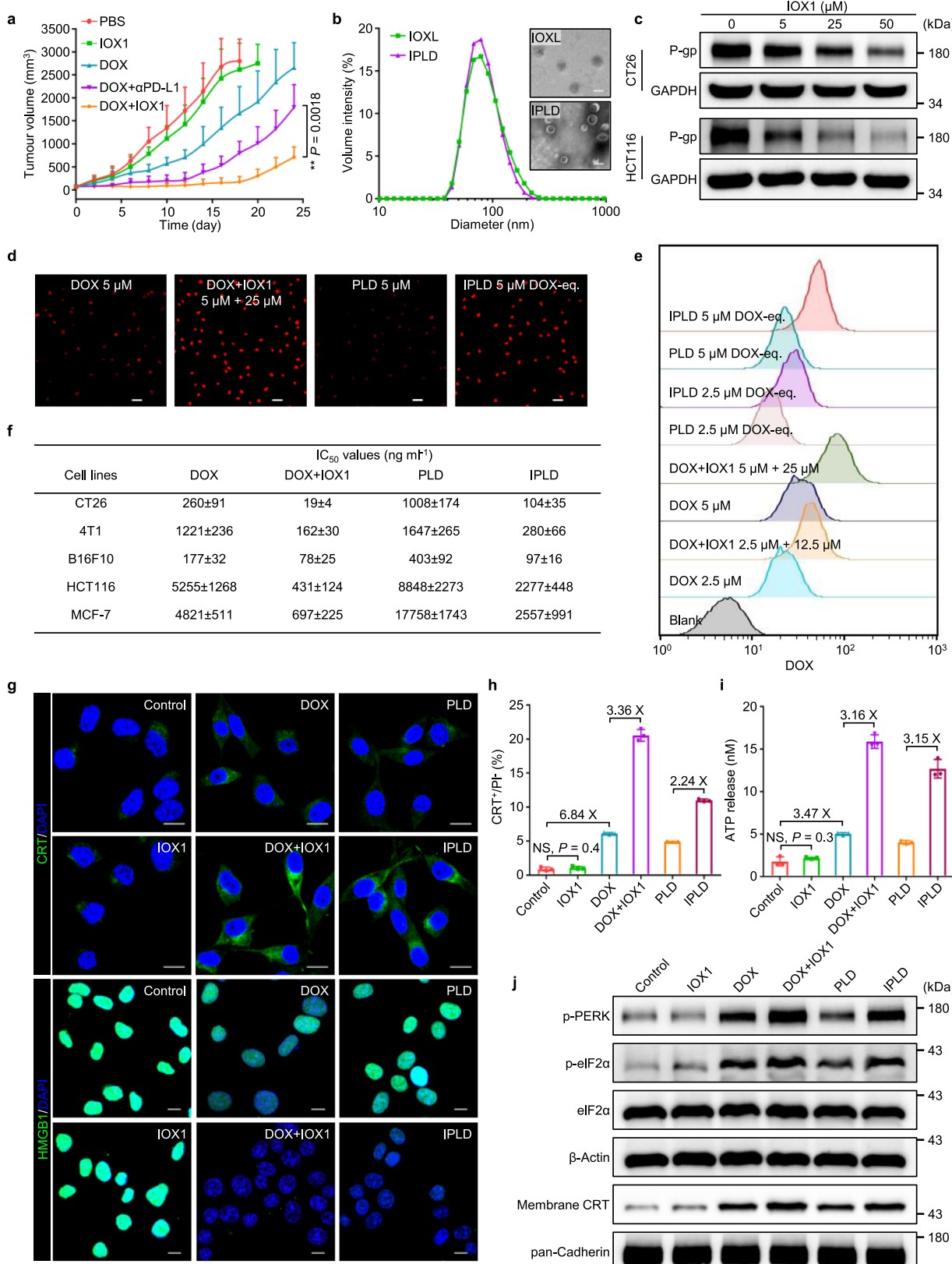

(Supplementary Figs. 28,29), the enhanced ICD induction by 5IOX + DOX is responsible for promoted DC maturation.

**IOX1 downregulates the PD-L1 expression of tumour cells.** PD-L1 overexpression of cancer cells is one of the major factors leading to cancer immunosuppressive microenvironment[1,7]. We found that IOX1 efficiently downregulated the membrane PD-L1 in CT26 cells in a concentration-dependent manner (Fig. 3d; Supplementary Fig. 30). The PD-L1 expression of CT26 cells was reduced from 33.0% in control to 14.5% by 5 μM IOX1 and even down to 9.9% by 25 μM IOX1. This downregulation was further confirmed by

**Fig. 2 IOX1 liposome formulation, inhibiting drug resistance, and potentiating cytotoxicity and ICD induction. a** Tumour growth curves of s.c. CT26 tumours in BALB/c mice treated via the tail vein injection of IOX1 (7.5 mg kg$^{-1}$), DOX (5 mg kg$^{-1}$), their combination (IOX1 + DOX) or DOX + αPD-L1 (DOX, 5 mg kg$^{-1}$, i.v.; αPD-L1,7.5 mg kg$^{-1}$, i.p.) on an every-two-day schedule for three times; initial tumour volume, 80 mm$^3$; $n$ = 6 mice. **b** Size distribution and morphology of liposomal IOX1 (IOXL) and IPLD (the mixed IOXL and PLD at an IOX1/DOX molar ratio of 5: 1) characterised by DLS and TEM; scale bars, 100 nm for the inset TEM images; $n$ = 3 independent samples. **c** Western blotting images of P-gp in CT26 and HCT116 cells treated with IOX1 (5, 25, and 50 μM) for 24 h. **d,e** Intracellular DOX concentration analysis by (**d**) the fluorescence images and (**e**) flow cytometry analysis of CT26 cells treated with DOX or its combination with IOX1 or their liposomes for 4 h; red: DOX; scale bars, 50 μm. **f** The IC$_{50}$ values of DOX, its combination with IOX1, or their liposomes against tumour cells; 48 h incubation; the IOX1/DOX molar ratio was kept at 5: 1; see Supplementary Fig. 16 for the dose-responsive curves. **g–j** ICD induction in CT26 cells treated with IOX1 or its combinations: (**g**) CLSM images of the CRT exposure (green) and the nuclear HMGB1 (green); blue: DAPI; scale bars, 10 μm; (**h**) flow cytometric analysis of CRT-positive cells (gated on the propidium iodide negative cells (PI-)) and (**i**) ATP secreted into the medium; $n$ = 3 independent experiments; (**j**) Western blotting images of p-PERK, p-eIF2α and membrane CRT; Cells were treated with IOX1 (25 μM), DOX (5 μM), their combination, or their liposomes for 4 h. In (**c–e,g,j**), the experiments were repeated independently three times to confirm the results. Data represent mean ± SD. Two-tailed Student's $t$ test. NS no significance; ** $P$ < 0.01. Source data are provided as a Source Data file.

western blotting (Fig. 3e). The downregulation was attributed to the reduction of its *Cd274* mRNA level as quantitated by qPCR (Fig. 3f). IOX1 could also downregulate the PD-L1 expression of HCT116 (Supplementary Figs. 31,32) and MCF-7 human cancer cells (Supplementary Fig. 33). IOXL exhibited the same effect as IOX1 with slightly less efficiency (Fig. 3d-f; Supplementary Fig. 34) due to the encapsulation.

Many chemotherapy drugs such as DOX can induce cancer cells to express more PD-L1[15,38,39]. Indeed, after treating with DOX or PLD, CT26 or HCT116 cells significantly upregulated their PD-L1 levels (Fig. 3d,e; Supplementary Figs. 30,31,35) resulting from the increased *Cd274* mRNA levels (Fig. 3f). IOX1 or its liposome reversed the DOX-induced PD-L1 overexpression and even reduced the PD-L1 and its mRNA levels lower than their controls (Fig. 3d-f; Supplementary Figs. 30,31,35).

**IOX1 promotes T cell proliferation and activity**. The PD-1/PD-L1 axis is a negative modulatory signaling pathway that triggers a variety of immunosuppressive effects, including T cell anergy; thus, blocking PD-1 or PD-L1 using antibodies[40,41] or down-regulating PD-L1 expression[42,43] was shown to promote the functions and proliferation of T cells. The influence of pre-treating tumour cells with IOX1 or its combination on T cell proliferation was investigated by measuring the reproduction ability of peripheral blood mononuclear cells (PBMCs) separated from BALB/c mice. PBMCs' reproduction activity was measured by a reported T cell proliferation index (PI), which is calculated from the reduction of the fluorescence intensity of carboxy-fluorescein succinimidyl ester (CFSE)[44,45]. CT26 cells were first treated with IOX1, DOX, 5IOX1 + DOX, or their liposomes for 24 h and then co-cultured with PBMCs in a fresh medium. As shown in Fig. 3g,h, the PBMCs with non-pretreated CT26 cells had a PI of 9.5%, while the PI of the PBMCs cultured with the 25 μM IOX1-pretreated CT26 cells increased to 25.8%. The DOX-treated CT26 cells also increased the PBMCs' PI to 22.5% because of the reduced number of cancer cells. Significantly, the cancer cells treated with 5IOX1 + DOX or IPLD allowed PBMCs to proliferate with the PI of 40.9% or 35.8% due to the IOX1-enhanced cytotoxicity and suppressed PD-L1 expression.

Subsequently, the apoptosis of tumour cells induced by PBMCs was analyzed (Fig. 3i,j). PBMCs had little ability to induce death of the control CT26 cells, while induced 14.5% IOX1-treated CT26 cells apoptotic. DOX alone caused 25.3% cell death by cytotoxicity, and the presence of PBMCs did not significantly enhance the cell apoptosis, indicating that even though DOX-treatment promoted PBMCs to proliferate by 22.5%, they were incompetent due to DOX-induced overexpression of PD-L1 on the cancer cells. The combination of 5IOX1 + DOX induced 51.4% cells apoptotic by cytotoxicity, while the presence of

PBMCs increased the population to 86.7%, suggesting 35.5% apoptosis elicited by PBMCs. A similar trend was found in the liposome formulation.

We also tested the influence of IOX1 or its combination-pretreated PD-L1-mutant tumour cells on T cell proliferation. IOX1 showed no effect on T cell proliferation with CT26 cells whose PD-L1 expression was knocked down (Supplementary Figs. 36,37) or overexpressed (Supplementary Figs. 36,38) due to its failure to regulate the PD-L1 expressions in PD-L1-mutant tumour cells. Importantly, different from normal CT26 cells, the PD-L1-overexpressing CT26 cells treated with 5IOX1 + DOX or IPLD could not influence the PBMCs' activity (Supplementary Fig. 39), indicating that IOX1 or its combination promoted the T cell proliferation and activity via, at least partially, suppressing the PD-L1 expression. Moreover, without cancer cells, IOX1 alone did not affect T cell proliferation (Supplementary Fig. 40).

**IOX1 regulates P-gp and PD-L1 expression by suppressing JMJD1A/β-catenin signaling**. IOX1 was reported to downregulate the β-catenin associated genes through inhibiting JMJD1 (KDM3)[46], mainly JMJD1A[47,48]. We knocked down JMJD1A using JMJD1A shRNA (sh*Jmjd1a*) to examine its relationship with P-gp and PD-L1 and their upstream signal β-catenin[49,50] (Fig. 4a). The results show that knockdown of JMJD1A in CT26 and HCT116 cells did significantly reduce the protein and mRNA levels of β-catenin (*Ctnnb1*), so did those of P-gp (*Abcb1*) and PD-L1 (*Cd274*) in both cell lines (Fig. 4b-e; Supplementary Figs. 41,42). On the other hand, adding *Ctnnb1* plasmid to JMJD1A-knockdown CT26 cells partially recovered their P-gp and PD-L1 expressions (Fig. 4f,g; Supplementary Fig. 43), further confirming that JMJD1A regulates P-gp and PD-L1 expression via the β-catenin signaling pathway (Fig. 4a). Furthermore, IOX1 could not affect the intracellular DOX accumulation (Supplementary Figs. 44,45) and protein expressions of P-gp and PD-L1 (Supplementary Figs. 46–49) anymore in JMJD1A-knockdown CT26 or HCT116 cells, confirming that JMJD1A was the main target of IOX1.

In the cells whose protein syntheses were blocked by cycloheximide, IOX1 sped up the reduction of the JMJD1A levels but did not affect its mRNA levels (Supplementary Fig. 50), indicating that IOX1 reducing JMJD1A protein levels (Fig. 4h,i; Supplementary Figs. 51,52) may be through promoting JMJD1A degradation. Thus, the above results indicate that IOX1 regulates P-gp and PD-L1 expression by β-catenin signaling transduction through not only inhibiting JMJD1A's demethylase activity[48] but also reducing its intracellular level in the cancer cells with JMJD1A overexpression. Therefore, IOX1 had no effect on normal cells (e.g. NIH-3T3, HUVEC) or immune cells having low JMJD1A levels (Supplementary Figs. 53–55).

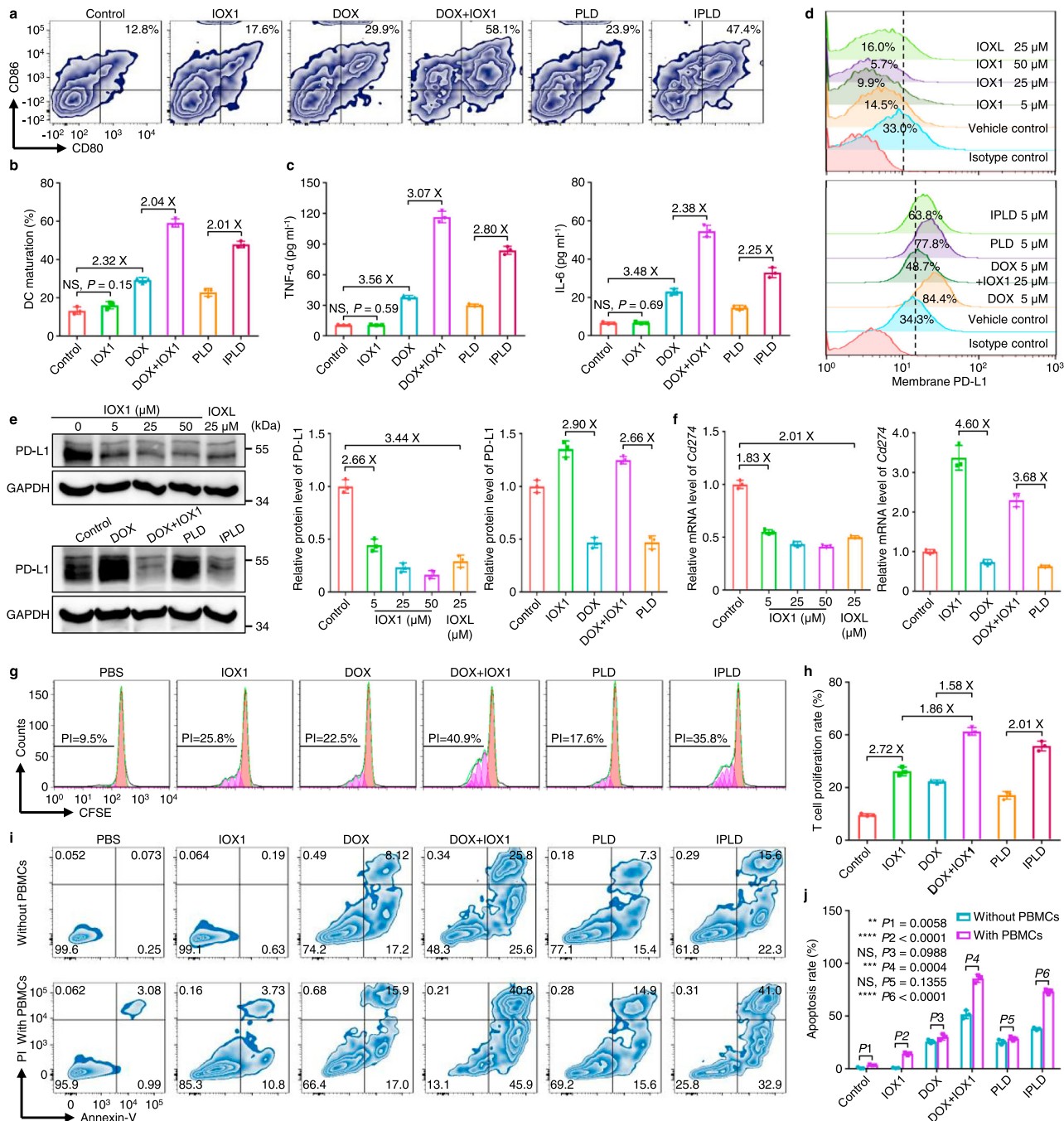

**Fig. 3 IOX1 mediates dendritic cell (DC) maturation, downregulates cancer cell PD-L1 expression, and promotes T cell proliferation and activity. a–c** DC maturation determined by (**a**) flow cytometry and (**b**) quantification of the mature DCs (CD11c+CD80+CD86+), and their secretion of (**c**) TNF-α and IL-6 in supernatants; CT26 cells were pre-treated with IOX1, DOX, their combination, PLD, or IPLD; DOX dose, 5 μM; IOX1 dose, 25 μM; 24 h; they were co-incubated with DCs for another 24 h; n = 3 independent experiments. **d–f** IOX1 downregulated the endogenous PD-L1 measured by (**d**) flow cytometry and (**e**) western blotting and quantification, and (**f**) the Cd274 mRNA levels analyzed by qPCR in the cells; CT26 cells were treated as indicated in (**a–c**) for 24 h; n = 3 independent experiments. **g–j** T cell (PBMC) proliferation and cytotoxic-activity after incubating with pre-treated CT26 cells: (**g,h**) flow cytometric profiles and their calculated proliferation indices (PIs) of T cells; **i,j** flow cytometric analysis and quantifications of the apoptotic CT26 cells with or without PBMCs co-culturing; CT26 cells were pre-treated as indicated in (**a–c**) for 24 h; the isolated cells were co-cultured with PBMCs in fresh medium for 3 days; the treated cells in fresh medium without PBMCs were cultured for comparison. n = 3 independent experiments. Data represent mean ± SD. Two-tailed Student's t test. NS no significance; ** P < 0.01, *** P < 0.001, **** P < 0.0001. Source data are provided as a Source Data file.

**The blood clearance and biodistribution of IOXL and IPLD.** The pharmacokinetics of IOXL and its mixed liposomes IPLD were very close to that of PLD, suggesting their excellent in vivo stability and slow clearance rates (Supplementary Fig. 56). The biodistribution in the s.c. CT26 tumour-bearing BALB/c mice showed that both IOXL and IPLD efficiently delivered more IOX1 to tumours than the free IOX1 by about eight times, and IPLD delivered DOX to the tumour as efficiently as PLD (Supplementary Fig. 57). The molar ratio of IOX1 to DOX in IPLD-treated tumours was about 4.93 : 1, close to the IPLD content

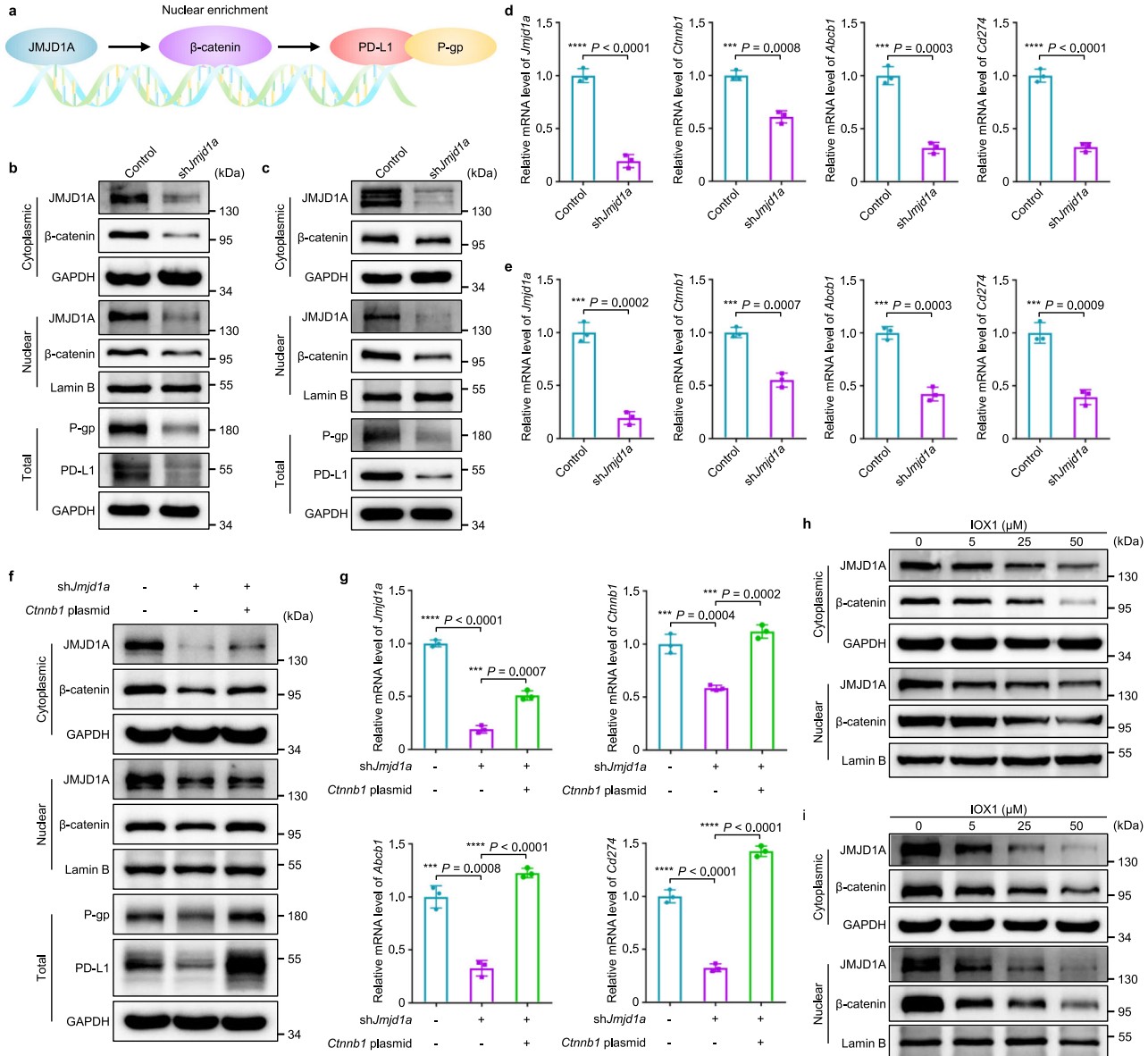

**Fig. 4 IOX1 downregulates P-gp and PD-L1 expression through JMJD1A/β-catenin signaling pathway. a** The pathway by which JMJD1A promotes P-gp and PD-L1 expression in tumour cells. The overexpressed JMJD1A increases the nuclear enrichment of β-catenin, which in turn accelerates the transcription and translation of downstream targets, including P-gp and PD-L1. **b–e** Effects of JMJD1A-knockdown on protein expressions measured by western blotting (**b,c**) or gene expressions measured by qPCR (**d,e**) in CT26 (**b,d**) or HCT116 (**c,e**) cells; $4 \times 10^5$ cells were transfected with sh*Jmjd1a* (2 μg/well) for 48 h; $n = 3$ independent experiments in (**d,e**). **f,g** Effects of β-catenin rescue in JMJD1A-knockdown CT26 cells on the P-gp and PD-L1 expressions: (**f**) the relative protein levels and (**g**) mRNA levels of JMJD1A (*Jmjd1a*), β-catenin (*Ctnnb1*), P-gp (*Abcb1*) and PD-L1 (*Cd274*) in CT26 cells; $4 \times 10^5$ cells were transfected with sh*Jmjd1a* (2 μg/well) for 48 h followed by transfection with *Ctnnb1* plasmid (2 μg/well) for 24 h; $n = 3$ independent experiments in (**g**). **h,i** IOX1 dose-dependent reduction of JMJD1A and β-catenin protein levels in CT26 (**h**) and HCT116 (**i**) cells; 24 h treatment. In (**b,c,f,h,i**), the experiments were repeated independently three times to confirm the results. Data represent mean ± SD. Two-tailed Student's $t$ test. ***$P < 0.001$, ****$P < 0.0001$. Source data are provided as a Source Data file.

(IOX1: DOX = 5: 1), further confirming the excellent in vivo stability of IPLD.

**IPLD eradicates small and large s.c. CT26 tumours.** The in vivo antitumour activity of the combination of 5IOX1 + DOX was evaluated using their mixed liposomes, IPLD, for their much improved pharmacokinetics and simplified injection. Small s.c. CT26 tumours of about 80 mm³ were first treated as scheduled in Fig. 5a at a DOX dose of 5 mg kg⁻¹, mostly used in literature. IOXL exhibited no significant tumour inhibition ability, and PLD

slowed down the tumour growth but regained growth momentum after the treatment withdrawal (Fig. 5b-d). However, IPLD treatments immediately made the tumours steadily regress, even after the third (last) injection until all the tumours were eradicated (complete response, CR) (Fig. 5b,c,e; Supplementary Figs. 58,59). After 7 days of the last treatment, three mice from each group were sacrificed, and the tumour sections were analyzed with the TUNEL-staining (Fig. 5f); the IPLD-treated tumours contained pervasive apoptotic cells, compared with some in the tumours treated with PLD and few in the tumours treated with IOXL-

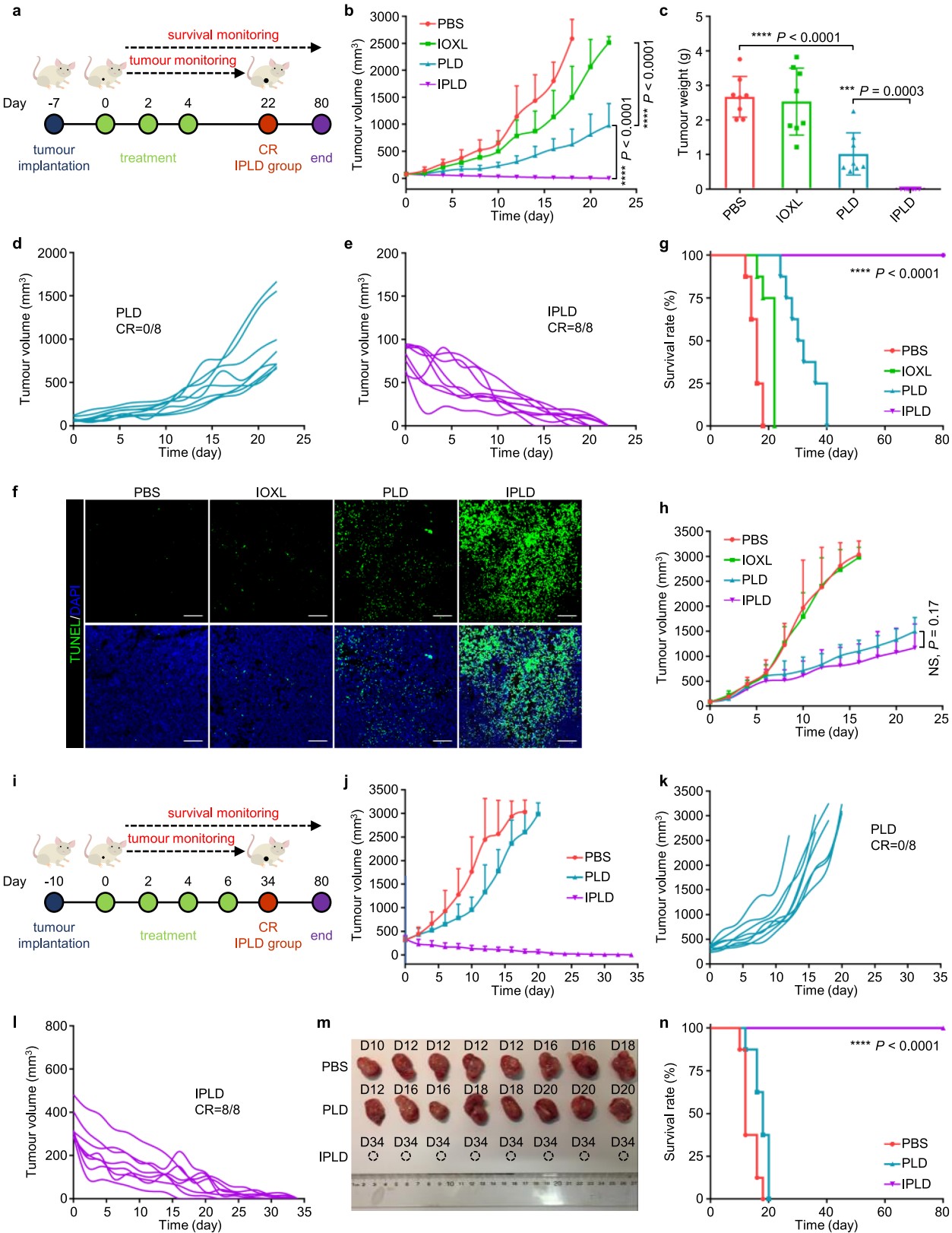

treated ones. All the IPLD-treated mice had no relapse and lived well through the 80 days observation (Fig. 5g; Supplementary Fig. 60); their body weights (Supplementary Fig. 83a) and the blood biochemistry were also unchanged compared with the controls (Supplementary Fig. 61), indicating that IPLD did not cause severe side effects.

Furthermore, a xenograft CT26 tumour model was used to assess the in vivo effect of JMJD1A-knockdown on tumour growth. Small s.c. JMJD1A-knockdown or control CT26 tumours of about 80 mm³ were treated as scheduled in Supplementary Fig. 62a at a DOX dose of 5 mg kg⁻¹. JMJD1A-knockdown tumours grew slightly slower than the control but all exceeded the

**Fig. 5 IPLD treatment of s.c. CT26 tumours. a–g** Treatment of small tumours: (**a**) Experimental schedule; After 7 days of s.c. inoculation with $5 \times 10^5$ CT26 cells, the tumours reached about 80 mm$^3$, and the treatment was initiated (noted as Day 0) with PBS, IOXL, PLD or IPLD at 7.5 mg kg$^{-1}$ IOX1, 5 mg kg$^{-1}$ DOX on every 2 days for 3 times. (**b**) The tumour growth curves, (**c**) the averaged tumour weight of each group on Day 22; $n = 8$ mice. (**d,e**) The individual tumour growth curves of the PLD group or the IPLD group in (**b**); CR: complete response. (**f**) TUNEL staining of the tumour sections dissected on day 11; green: TUNEL; blue: DAPI; scale bars: 100 μm; the experiment was repeated independently three times to confirm the results. (**g**) Kaplan–Meier analysis of the mice; $n = 8$ mice. **h** The tumour growth curves of s.c. CT26 tumours in nude mice treated as in (**a**); $n = 6$ mice. **i–n** Treatment of large s.c. 350 mm$^3$ CT26 tumours in BALB/c mice: (**i**) Experimental schedule; treatments as in (**a**) every 2 days for 4 times. (**j**) The tumour growth curves; $n = 8$ mice. (**k,l**) The individual tumour growth curves of the PLD group or the IPLD group. (**m**) The images of the tumours dissected on the indicated days. (**n**) Kaplan–Meier analysis of the mice; $n = 8$ mice. Data represent mean ± SD. Statistical significance was analyzed by two-tailed Student's $t$ test (**b,c,h**) or by the log-rank (Mantel–Cox) test (**g,n**). NS no significance; *** $P < 0.001$, **** $P < 0.0001$. Source data are provided as a Source Data file.

maximum allowable volume after 22 days (Supplementary Fig. 62b,e,f). PLD slowed down the growth of control tumours as the previous results (Supplementary Fig. 62b,c,e,f). Notably, JMJD1A-knockdown enabled PLD to realise the tumour eradication (Supplementary Figs. 62b,d-f & 63), similar to the IOXL treatment. Western blotting analysis showed that JMJD1A-knockdown tumours had lower protein levels of JMJD1A and its downstream targets, β-catenin, P-gp and PD-L1, again similar to the IOXL-treated ones (Supplementary Fig. 64).

Interestingly, when treating CT26 tumours in congenital immunocompromised T-cell-deficient athymic nude mice, IPLD showed no difference from PLD (Fig. 5h; Supplementary Figs. 65, 66), proving the critical immunotherapy role of IOX1 in the enhanced antitumour activity of IPLD.

Large tumours generally respond poorly to various therapies, leading to poor survival due to the enhanced immunosuppressive microenvironment[12]. We further challenged IPLD with tumours of ~350 mm$^3$ as an inoperable model (Fig. 5i). Although PLD slowed down small tumours' growth, it did little to the large tumours (Fig. 5j,k). In contrast, IPLD at the same dose of DOX shrank the tumours, even after the first treatment, and finally eradicated the tumours in the 4 weeks post-treatment (Fig. 5j,l,m; Supplementary Fig. 67). No relapse was observed in the follow-up observation, and all the mice were very healthy even at the end of the experiment, 80 days after the first treatment (Fig. 5n; Supplementary Fig. 68).

After three treatments with the indicated formulations, the tumours were dissected for immunofluorescence analysis (Fig. 6a). The tumours treated with IPLD had the highest CRT signal intensity, 3.0-fold of the PLD-treated tumours (Fig. 6b), while their PD-L1 expression was just 1/3 of the PLD-treated tumours and even lower than the PBS group (Fig. 6c). Notably, IOXL downregulated the PD-L1 expression to a lower level compared with the PBS and IPLD groups but had no antitumour activity.

The tumour-draining lymph nodes (TDLNs) of the treated mice were collected to investigate the DC maturation by flow cytometric analysis of CD80$^+$ and CD86$^+$, two biomarkers of mature DCs[51]. The IPLD-treated mice had as high as 53.6% CD80$^+$CD86$^+$ mDCs in TDLNs compared with 24.6% in the PLD-treated mice and 11.0% in the PBS group (Fig. 6d,e), promoting the recruitment of T cells into the tumours (Supplementary Fig. 69).

IPLD-treatment also recruited 62.2% CD8$^+$ T cells into the tumours, 1.3-fold of that in the PLD-treated tumours (Fig. 6f,g). CD8$^+$ T cells, termed as cytotoxic T lymphocytes (CTLs), attack cancer cells by secreting an apoptosis trigger, granzyme B (GB)[52]. The immunofluorescence of CTLs and GB in the tumour tissues showed that IPLD-treated tumours did have much higher contents of CD8$^+$ CTLs and GB (Fig. 6h), which were 4.7-fold (Fig. 6i) and 4.1-fold (Fig. 6j) of those in the PLD-treated tumours. However, the population of immunosuppressive regulatory T cells (Tregs) in the IPLD-treated tumours was the lowest, only about half

of that in the PLD or IOXL group and 1/3 of the control (Fig. 6k,l).

The lymphocytes collected from the spleens were further analyzed to probe the IPLD-triggered activation of the systemic antitumour immune response. The proportion of CTLs was remarkably increased (Fig. 6m; Supplementary Fig. 70) while the population of Tregs (CD4$^+$FOXP3$^+$) was reduced (Supplementary Fig. 71) in the IPLD-treated mice in comparison with those of the PBS, IOXL, or PLD groups.

The systemic concentrations of circulating cytokines, including IFN-γ, TNF-α, IL-2, IL-6, and IL-10 in the mice's serum, were determined (Fig. 6n). The IPLD-treated mice had significantly higher levels of IFN-γ, TNF-α, IL-2, IL-6 than the mice in the PLD and other groups, while the IL-10 levels were unchanged.

The CTLA-4 expression of intratumoural Tregs (Supplementary Fig. 72) and the population of immuosuppresive tumour-associated M2 macrophage (Supplementary Fig. 73) in the IPLD-treated mice did not change after 7 days of the last treatment, indicating the important role of CTLs in tumour inhibition.

One serious side effect of the PD-1/PD-L1 checkpoint blockade therapies using antibodies is inducing type-1 diabetes because T cells also attack the pancreatic cells whose PD-L1 is blocked[53,54]. Immune-mediated pneumonitis caused by undifferentiated aggressiveness of immune cells is another potential risk of immune-checkpoint inhibitors[55]. Fortunately, IOXL and IPLD treatments did not influence the PD-L1 expression in the pancreas (Supplementary Fig. 74) and lungs (Supplementary Fig. 75). In addition, IOXL did not affect while PLD and IPLD slightly increased the CTLs' PD-L1 levels (Supplementary Fig. 76).

**IPLD elicits long-term antitumour immunological memory.** The long-term immunological memory was tested by rechallenging the treated BALB/c mice bearing s.c. 80 mm$^3$ CT26 tumours (Fig. 7a). On 3 days after the last treatment, as indicated above, the tumours in the mice of the PBS, IOXL and PLD groups were surgically removed; the IPLD-treated mice were all CR, so needed no operation. The mice were rechallenged with CT26 cells via s.c. injection on Day 30. The rechallenging cells in the mice pre-treated with PBS formed tumours growing rapidly as the first inoculation (Fig. 7b). In the mice treated with IOXL or PLD, the tumour formation delayed a few days but later grew similarly to the control. Notably, all the IPLD-treated mice completely rejected tumours and survived well (Fig. 7b,c; Supplementary Fig. 77).

The IPLD-induced immunological memory was further demonstrated with a lung metastasis tumour model by i.v. injecting luciferase-expressing CT26 ($^{Luci}$CT26) tumour cells into the pre-treated BALB/c mice as described above (Fig. 7a; Supplementary Fig. 78). As monitored by bioluminescence, the i.v. inoculated cells quickly established lung tumours in the mice pre-treated with PBS, IOXL or PLD and killed them quickly (Fig. 7e). No bioluminescence was found in the IPLD-pretreated

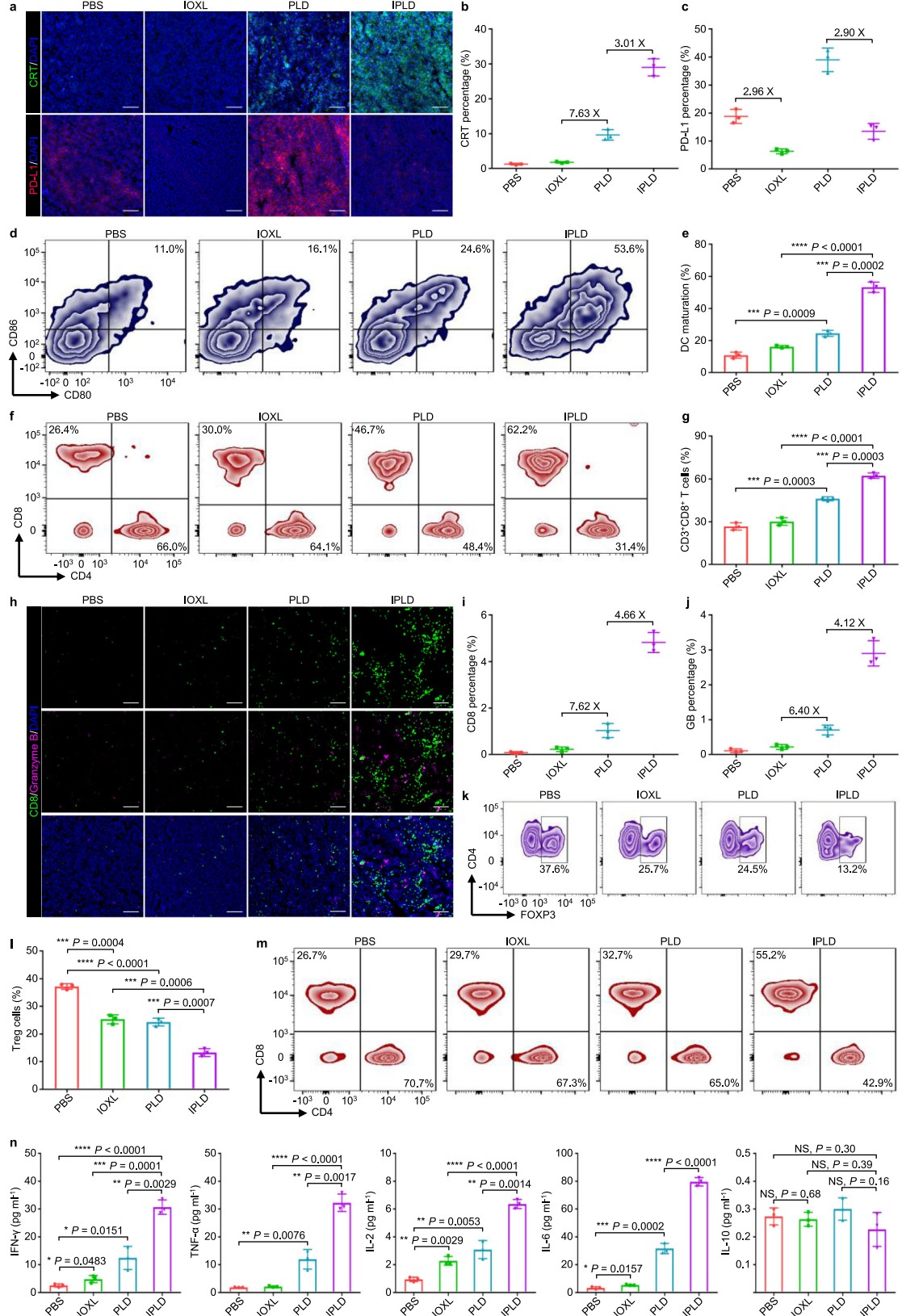

mice during the 1-month monitoring (Fig. 7e), and even 2 months after the i.v. inoculation (Supplementary Fig. 79), suggesting rejection of tumour formation, and thus all the mice lived healthily until the end of the experiment.

These results demonstrate that treatment with IPLD produced tumour-specific immunological memory. Analysis of the central

memory T cells ($T_{CM}$, CD4$^+$CD44$^+$ CD122$^+$) in the spleens in the primary CT26 tumour-bearing mice on day 11 (1 week after the third (last) treatment) showed that the spleens of the mice treated with IPLD had a significantly increased population of $T_{CM}$ compared with those treated with PBS, IOXL, or PLD (Fig. 7d; Supplementary Fig. 80). Effector memory T cells ($T_{EM}$) are

**Fig. 6 IPLD-induced immune responses of the CT26 tumour-bearing BALB/c mice.** The mice were inoculated and treated as the schedule in Fig. 5a; on 7 days after the last treatment (Day 11), the serum, tumours, tumour-draining lymph nodes (TDLNs), and spleens were collected for analysis. **a–c** CRT exposure and PD-L1 expression in the tumours by (**a**) immunostaining (green: CRT; red: PD-L1; blue: DAPI; scale bars, 100 μm) and the quantification of (**b**) CRT exposure and (**c**) PD-L1 expression by ImageJ; $n = 3$ mice. **d,e** Flow cytometry analysis of the mature DCs (CD11c$^+$CD80$^+$CD86$^+$) in TDLNs: (**d**) the contour diagrams and (**e**) quantification; $n = 3$ mice. **f,g** Flow cytometry analysis of cytotoxic T lymphocytes (gated on CD3$^+$CD8$^+$, CTLs) in the tumours: (**f**) the contour diagrams and (**g**) quantification; $n = 3$ mice. **h–j** CD8 and granzyme B in the tumours: (**h**) immunostaining images (green: CD8; pink: granzyme B; blue: DAPI; scale bars, 100 μm) and the quantification of (**i**) CD8 and (**j**) granzyme B expression by ImageJ; $n = 3$ mice. **k,l** Flow cytometry analysis of regulatory T cells (CD4$^+$FOXP3$^+$) in the tumours: (**k**) the contour diagrams and (**l**) quantification; $n = 3$ mice. **m** Flow cytometry analysis of the CTLs (CD3$^+$CD8$^+$) in the spleens; the experiment was repeated independently three times to confirm the results. **n** Quantification of systemic cytokines in the serum of the mice; $n = 3$ mice. Data represent mean ± SD. Two-tailed Student's $t$ test. NS no significance; *$P < 0.05$, **$P < 0.01$, ***$P < 0.001$, ****$P < 0.0001$. Source data are provided as a Source Data file.

known to induce immediate protections by producing multiple cytokines, including IFN-γ and TNF-α, upon reencountering the antigens[56]. The immune memory related cytokines in the mice's serum on Day 6, 29, and 32 (just before or after tumour cell rechallenge) were analyzed (Fig. 7f,g). The IPLD treatment induced high levels of both cytokines in the serum (day 6), and both returned to the normal levels on day 29; their serum levels were significantly elevated again once rechallenged with CT26 cells, verifying IPLD-treatment did activate $T_{EM}$ in the mice. In contrast, the cytokine levels in the PLD- and IOXL-treated mice did not respond well.

**IPLD cures 4T1 orthotopic and lung metastatic dual tumours.** IPLD was further challenged to treat late-stage cancer model, the mice bearing both orthotopic primary and metastatic tumours of less immunogenic triple-negative 4T1 breast cancer[57]. For the dual tumour model, the mice bearing 100 mm$^3$ orthotopic tumours were first treated with three i.v. injections; three days after the last treatment, the mice were i.v. injected with $^{Luci}$4T1 cells as circulating tumour cells (CTCs) (Fig. 8a). As expected, the IPLD treatments enhanced the mice's immune response, as evidenced by the high serum immune cytokines analyzed on Day 9 compared with the PLD-treatment group (Fig. 8b).

As usual, the PLD treatment alone slightly slowed down the primary orthotopic tumour growth (Fig. 8c; Supplementary Fig. 81). IPLD treatments again shrank the primary tumours, three of which achieved CR, one tumour did not grow, and one initially diminished but rebounded on Day 10 (Fig. 8d), but both tumours at the end of the experiment were much smaller than those in other groups (Fig. 8e). The in vivo bioluminescence tracking of lung $^{Luci}$4T1 tumours showed that the PBS, IOXL, and PLD-treated groups had strong bioluminescence in consistence with the massive numbers of large and small tumours in their lungs (Fig. 8f,g; Supplementary Fig. 82). In contrast, the three primary tumour-free mice in the IPLD-treated group had no bioluminescence signal at all (Fig. 8f), and no tumour was found in their dissected lungs (Fig. 8g); the other two mice bearing small primary tumours had barely observable bioluminescence, and there were only three tiny nodules in the two lungs (Fig. 8f,g).

## Discussion
The combination of chemotherapy and immunotherapy, especially with immune checkpoint inhibitors, has been shown to enhance the outcomes of many cancers[10,58–60]. The rationale is that some chemotherapeutic drugs turn non-immunogenic tumour cells to immunogenic, i.e. ICD[13,16,61], to present antigens to DCs for recruiting T cells, while anti-PD-1/PD-L1 antibodies disrupt the negative immune-regulation and thus normalise the T cell functions[10,59].

Herein, we show that IOX1 enabled a paradigm of potent antibody-independent chemo-immunotherapy, where IOX1 effectively downregulated the PD-L1 expression of cancer cells and simultaneously stimulated the DOX-induced ICD through deactivation of the JMJD1A/β-catenin pathway (Fig. 1).

IOX1 significantly increased the intracellular DOX concentration in tumour cells by suppressing the P-gp expression and inhibiting its activity, thus enhancing DOX's cytotoxicity (Fig. 2c-f). The increased intracellular DOX concentration also induced strong ICD characterised by significant CRT exposure (Fig. 2g,h,j), HMGB1 release (Fig. 2g), and ATP secretion (Fig. 2i). These signal molecules known as DAMPs stimulated DC maturation, as seen that the 5IOX1 + DOX-treated CT26 cells promoted DCs' maturation nearly two times as much as the DOX-treated cells (Fig. 3a,b). Accordingly, the BMDCs cultured with 5IOX1 + DOX-treated cancer cells released high levels of TNF-α and IL-6 (Fig. 3c) as tumour antigens stimulated mDCs to secrete pro-inflammatory cytokines[51,62]. The mDCs facilitated the recruitment and infiltration of T cells into the tumour tissue (Supplementary Fig. 69).

IOX1 effectively downregulated the PD-L1 expression of cancer cells in a concentration-dependent manner via the suppression of JMJD1A/β-catenin/PD-L1 pathway (Fig. 3d-f; Fig. 4h,i). Importantly, IOX1 was able to reverse the DOX-induced PD-L1 overexpression of cancer cells. It is known that many chemotherapeutic drugs can promote cancer cells to upregulate PD-L1[38,63,64]. For instance, DOX treatment stimulated cancer cells to express 2.5 times more PD-L1 (Fig. 3d), enabling them to escape immune monitoring more efficiently[3]. The presence of IOX1 made the DOX-treated cells downregulate their PD-L1 expression even lower than the controls (Fig. 3d-f). Therefore, IOX1 disrupts the PD-1/PD-L1 axis between T cells and cancer cells. Indeed, the T cells co-cultured with the tumour cells pre-treated with 5IOX1 + DOX or IPLD were more proliferative (Fig. 3g,h) and more active (cytotoxic) (Fig. 3i,j). The dual effects of IOX1, enhancing DOX-induced ICD and downregulating tumour PD-L1, effectuated potent antitumour efficacy of DOX + IOX1 combination, significantly higher than the DOX + αPD-L1 combination (Fig. 2a).

For in vivo applications, IOX1 was loaded into the PLD-like liposome, IOXL, for improved pharmacokinetics and tumour accumulation (Supplementary Figs. 56,57). IOXL was further mixed with PLD at an IOX1 to DOX molar ratio of 5: 1 to obtain the mixed liposomes (Fig. 2b), IPLD, for injection convenience. In vitro experiments showed that IOXL, PLD, and IPLD retained their biological activities as small molecule drugs (Figs. 2,3).

IPLD demonstrated high antitumour efficacy. At 5 mg kg$^{-1}$ of DOX and 7.5 mg kg$^{-1}$ of IOX1, IPLD enabled all the mice to achieve CR in not only small (80 mm$^3$) but also incurable exponentially-growing large (350 mm$^3$) CT26 colon tumour models (Fig. 5b-e,j-m). Indeed, IPLD-treated mice had substantially higher CRT exposure, lower tumour PD-L1 expression

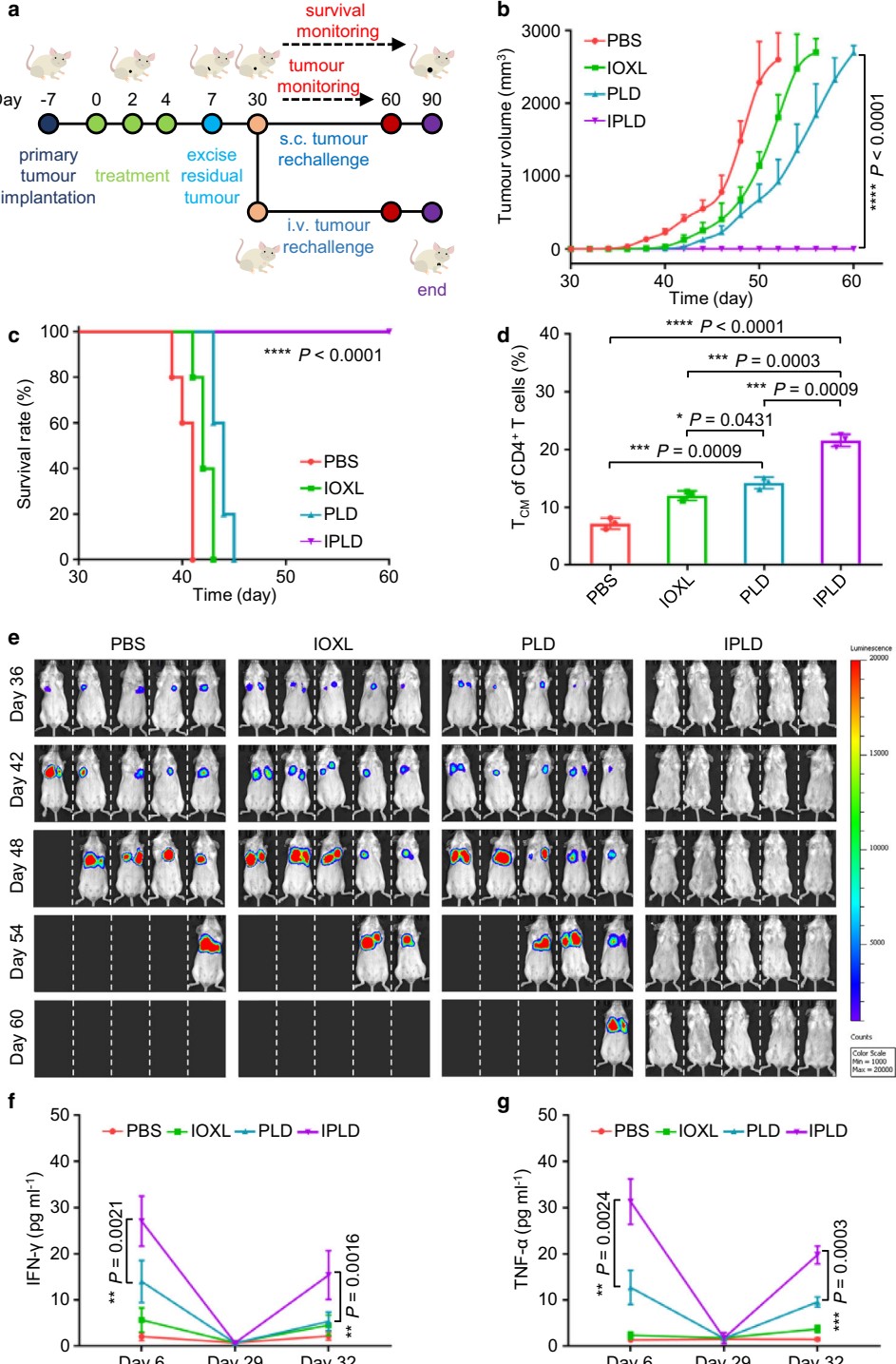

**Fig. 7 IPLD treatment-induced long-term antitumour immunological memory tested by tumour rechallenging. a** Experimental schedule: BALB/c mice with primary s.c. CT26 tumours of about 80 mm³ were treated with the liposome formulas at 7.5 mg kg⁻¹ IOX1, 5 mg kg⁻¹ DOX every 2 days for three times; 3 days after the last treatment (Day 7), the tumours of the mice in the PBS, IOXL and PLD groups were surgically removed, and the wounds were sutured; the IPLD group needed no surgery as all the primary tumours disappeared after treatment. On Day 30, the mice in each treatment group were divided into two subgroups; one was s.c. inoculated with CT26 cells (5 × 10⁵) and the other was i.v. injected with 5 × 10⁵ ᴸᵘᶜⁱCT26 cells. **b,c** The tumour growth curves (**b**) and Kaplan–Meier analysis (**c**) of the mice bearing the rechallenged s.c. tumours; n = 5 mice. **d** Quantitative analysis of the central memory T cells (CD4⁺CD44⁺CD122⁺) in spleens of the mice on Day 11; n = 3 mice. **e** In vivo bioluminescence imaging the rechallenged lung tumours at different times; n = 5 mice. **f,g** The IFN-γ and TNF-α levels in the serum of mice in (**e**) on Day 6, 29, and 32; n = 3 mice. Data represent mean ± SD. Statistical significance was analyzed by two-tailed Student's *t* test (**b,d,f,g**) or by the log-rank (Mantel–Cox) test (**c**). * P < 0.05, ** P < 0.01, *** P < 0.001, **** P < 0.0001. Source data are provided as a Source Data file.

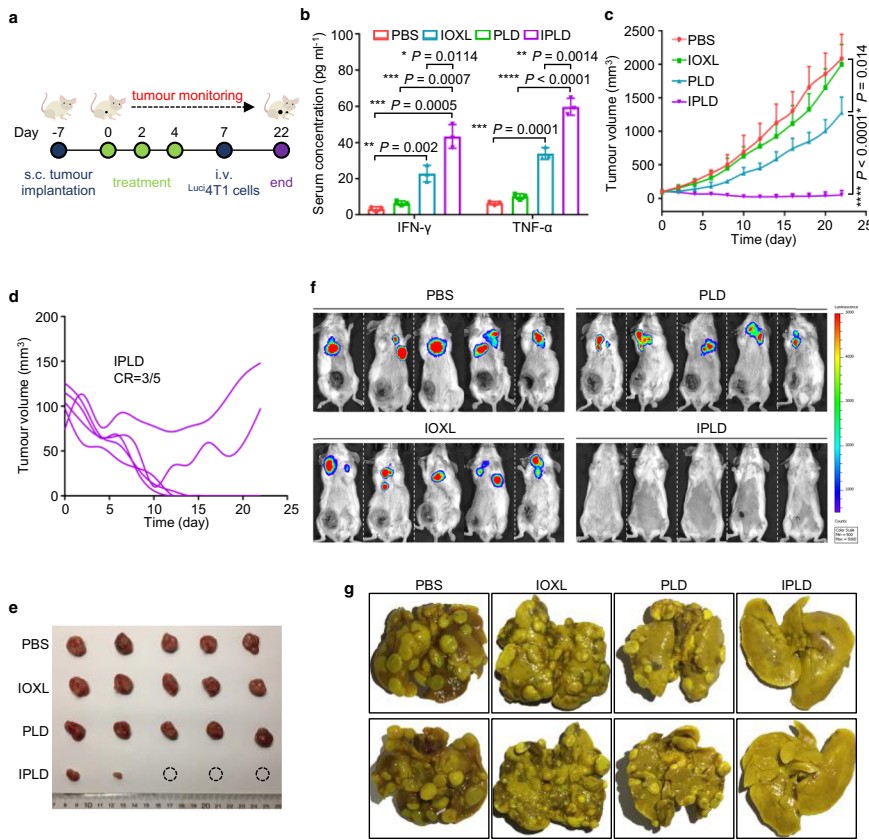

**Fig. 8 IPLD treatment of orthotopic and lung metastatic dual tumours of 4T1 triple-negative breast cancer. a** Experimental schedule for the dual tumour model: BALB/c mice were s.c. inoculated with $1 \times 10^6$ 4T1 cells at the left mammary gland; when the tumours reached about 100 mm$^3$, the treatments were initiated (noted as Day 0) with the liposome formulas at 7.5 mg kg$^{-1}$ IOX1, 5 mg kg$^{-1}$ DOX or their combination every 2 days for three times; 3 days after the last treatment (Day 7), $^{Luci}$4T1 cells ($5 \times 10^5$) were injected to the mice via the tail vein. **b** The serum IFN-γ and TNF-α levels in the mice determined on day 9; $n = 3$ mice. **c** The tumour growth curves of the orthotopic 4T1 tumours; $n = 5$ mice. **d** The individual tumour growth of each mouse in the IPLD-treated group; CR: complete response. **e** The images of the tumours dissected at the end of the experiment (Day 22). **f** In vivo live bioluminescence imaging the $^{Luci}$4T1 tumours in lungs on day 22; $n = 5$ mice. **g** Representative photographs of the lungs dissected from the mice on day 22. Data represent mean ± SD. Two-tailed Student's $t$ test. * $P < 0.05$, ** $P < 0.01$, *** $P < 0.001$, **** $P < 0.0001$. Source data are provided as a Source Data file.

(Fig. 6a–c), and more mDCs (Fig. 6d,e) and active CTLs (Fig. 6f,g) but less Tregs (Fig. 6k,l), as well as higher levels of antitumour-associated cytokines including IFN-γ, TNF-α, IL-2, and IL-6 (Fig. 6n) but unchanged anti-inflammatory cytokine IL-10 (known detrimental to antitumour immune response[65]). All CR in immunocompetent mice compared with the partial response in immunocompromised nude mice after the same IPLD treatment clearly reflected the contribution of the IOX1-induced immune responses to antitumour activity (Fig. 5h).

Moreover, the IPLD treatment produced long-term memory of antitumour immunities, as evidenced by the increased numbers of T$_{CM}$ (Fig. 7d) and the elevated cytokines from T$_{EM}$ upon tumour rechallenging (Fig. 7f,g). T$_{CM}$ residing in the secondary lymphoid tissues provide immunities after expansion, differentiation, and migration, while T$_{EM}$ locating in both lymphoid and non-lymphoid tissues provide immediate protections by producing multiple cytokines, including IFN-γ and TNF-α upon reencountering the antigens[56]. The serum levels of IFN-γ and TNF-α increased significantly after IPLD treatment and returned to the normal levels after healing (Fig. 7f,g), but increased greatly again upon the second inoculation and enabled mice to reject the rechallenged tumours (Fig. 7a–c,e).

The strong immune activation ability of IPLD was further demonstrated in challenging 4T1 orthotopic and lung metastatic dual tumour model, mimicking late-stage cancer. Similarly, three IPLD-treatments achieved 3 CR out of the 5 with primary orthotopic (Fig. 8d,e) and the distant lung metastatic tumours (Fig. 8f,g), as resulted from the activated antitumour immunity[66]. The left 2 mice only partially responded, but their primary and metastasis tumours were far smaller and less (in the lung) than those of the PLD group.

IPLD did not show apparent adverse effects during the treatments. IOX1 did not amplify DOX's cytotoxicity to non-cancer cells such as NIH-3T3 and HUVEC cells due to their low JMJD1A levels (Supplementary Figs. 17,53,54). The treatments also caused no obvious changes in mice's body weights (Supplementary Fig. 83) and blood chemistry (Supplementary Fig. 61). The mice's body weights with large tumours dropped slightly at the beginning of the IPLD treatment due to the rapid tumour regression but stayed stable later (Supplementary Fig. 83b).

The undifferentiated aggressiveness is a major limitation of PD-1/PD-L1 checkpoint blockade therapies[26]. For instance, pancreatic cells express PD-L1 to protect themselves from the attack by activated lymphocytes through the PD-1/PD-L1 pathway[54]; thus, the nonspecific binding by anti-PD-L1 antibody increased the risk of autoimmune diabetes in patients[53]. Pneumonitis is also a known, potentially fatal toxicity of anti-PD-1/PD-L1 immune checkpoint inhibitors[55]. Moreover, several studies reported that PD-L1 deficient CTLs exhibiting increased contraction were susceptible to being killed by other CTLs[67]; therefore, the PD-1/PD-L1 checkpoint blockade therapies deteriorated the functions of CTLs[68]. Interestingly, IOXL, as well as IPLD, did not affect the PD-L1 expression of

the low JMJD1A-expressing pancreas[69] (Supplementary Fig. 74), lungs[70] (Supplementary Fig. 75) and CTLs (Supplementary Fig. 76). Thus, IOX1 may avoid these complications of antibody-therapy. Indeed, all the IPLD-treated mice did not show these complications and lived well through the experiments.

In conclusion, IOX1 effectively downregulates the PD-L1 expression in cancer cells and even reverse the DOX-induced high PD-L1 expression, disrupting the PD-1/PD-L1 axis and restoring the proliferation and activity of T cells. IOX1 also substantially enhanced the DOX-induced ICD of cancer cells, and thus efficiently remodeled the tumour-immunosuppressive microenvironment, including more DC maturation, T cell infiltration and accordingly the antitumour cytokines. Therefore, the liposomal formulation of 5IOX1 + DOX generated not only immediate antitumour immune responses that cured the primary s.c. tumours including orthotopic 4T1 tumours but also long-term antitumour memory that rejected tumour rechallenge. Furthermore, as JMJD1A is much more overexpressed in cancerous tissues than in healthy tissues[70], the IOX1 therapy may cause few complications compared with the antibody-therapy. Thus, IOX1 may enable a highly potent antibody-free chemo-immunotherapy for cancers.

## Methods

**Materials.** 5-Carboxy-8-hydroxyquinoline (IOX1), doxorubicin hydrochloride (DOX), cholesterol, daunorubicin (DNR), epirubicin (EPI), and oxaliplatin (OXA) were purchased from Sigma-Aldrich Co. (St. Louis, USA). 1, 2-Dioleoyl-sn-glycero-3-phosphoethanolamine (DOPE), 1, 2-distearoyl-sn-glycero-3-phosphoethanol-amine-N-[methoxy (polyethylene glycol)-2000] (DSPE-mPEG 2000) were purchased from Avanti Polar Lipids Inc. (Alabaster, USA). Pegylated liposomal DOX (PLD, LIBOD®) was obtained from the First Affiliated Hospital of Zhejiang University School of Medicine. Dulbecco's Modified Eagle's Medium (DMEM) and Roswell Park Memorial Institute-1640 (RPMI-1640) cell culture medium and fetal bovine serum (FBS) were purchased from Thermo Fisher Scientific Inc. (Waltham, USA). 3-(4, 5-Dimethylthiazol-2-yl)-2, 5-diphenyltetrazolium bromide (MTT) and 5, 6-carboxy-fluorescein diacetates succinimidyl ester (CFSE) were obtained from Dalian Meilun Biotech. Inc. (Dalian, China). Anti-JMJD1A, anti-β-Catenin, anti-PD-L1, anti-P-gp, anti-GAPDH, anti-β-Actin, anti-pan-Cadherin, anti-LC3B, anti-CRT and anti-HMGB1 antibodies were purchased from Abcam Co. (Cambridge, UK). Anti-p-PERK, anti-eIF2α and anti-p-eIF2α antibodies were purchased from Cell Signalling Technology (Danvers, USA). 4′, 6-Diamidino-2-phenylindole (DAPI), Alexa Fluor 488-labeled goat anti-rabbit IgG and Alexa Fluor 647-labeled goat anti-rabbit IgG, membrane and cytosol protein extraction kit, ATP detection kit, Annexin V-FITC/PI apoptosis detection kit and one step TUNEL apoptosis assay kit were obtained from Beyotime Inc. (Shanghai, China). Mouse IFN-γ, TNF-α, IL-2, IL-6, and IL-10 ELISA kits were purchased from Dakewe Biotech. Inc. (Shanghai, China). FITC anti-CD11c, APC anti-CD86, PE anti-CD80, PerCP/Cy5.5 anti-CD44, APC anti-CD122, FITC anti-CD45, APC anti-F4/80, PE-Cy7 anti-CD206 and PE-Cy7 anti-CD152 antibodies were purchased from BioLegend Inc. (San Diego, USA). APC anti-CD3, PE anti-CD4, FITC anti-CD8α and Alexa Fluor 647 anti-FOXP3 antibodies were purchased from BD Co. (Franklin Lakes, USA).

**Cell lines.** Murine colon cancer cell line CT26 (ATCC® CRL-2638), murine breast cancer cell line 4T1 (ATCC® CRL-3406), murine melanoma cell line B16F10 (ATCC® CRL-6475), murine embryonic fibroblast cell line NIH-3T3 (ATCC® CRL-1658), human colon cancer cell line HCT116 (ATCC® CCL-247), human breast cancer cell line MCF-7 (ATCC® HTB-22) and human umbilical vein endothelial cell line HUVEC (ATCC® CRL-1730) were purchased from ATCC (American Type Culture Collection). They were maintained in RPMI 1640 or DMEM supplemented with 10% FBS and 1% penicillin-streptomycin solution. Luciferase expressing ^Luci^CT26 and ^Luci^4T1 cells were established by stably transfection with pLV-Luci (Genechem, Shanghai, China) according to the manufacture's instruction. All the cells were cultured at 37 °C in 5.0% CO₂. The cells were checked for mycoplasma contamination and then resuspended for use.

**RNAi and gene transfection.** The short hairpin RNA (shRNA) targeting human Jmjd1a, murine Jmjd1a, murine Cd274, murine Abcb1 or murine Calr cloned into LV3-puro vectors, and pEX-3 vectors containing overexpression plasmid of murine Ctnnb1, murine Cd274, or murine Abcb1 were provided by GenePharma (Shanghai, China). Transfection with shRNA or overexpression plasmid was performed with Lipofectamine™ 2000 (Invitrogen, Carlsbad, USA) according to the manufacturer's instructions. Typically, cells (4 × 10⁵ cells/well) were seeded in 6-well plates and the plasmid (2 μg) was added when the cells reached 50% confluence.

After 6 h transfection, the supernatant was replaced by a complete culture medium. The cells were collected after 24 h for qPCR or 48 h for western blotting analysis. Puromycin (2 μg ml⁻¹) was utilised to screen out the stable knockdown cell lines. The shRNA specific targeting sequences (5′-3′) are listed as follows: human Jmjd1a, AGAAGAATTCAAGAGATTCCGGAGG; murine Jmjd1a, GTACAAGAAGCAG TAATAA; murine Cd274, GCGTTGAAGATACAAGCTCAA; murine Abcb1, CC AGCATTCTCCGTAATAT; murine Calr, CCGCTGGGTCGAATCAA.

**Animals.** BALB/c mice (6 weeks, female) and BALB/c nude mice (6 weeks, female) were purchased from the Institute of Medicine of Zhejiang Province. All mice were housed under a 12 h light-dark diurnal cycle with controlled temperature (20–25 °C) and relative humidity (40–60%). Food and water were available ad libitum. Experiments were carried out under pathogen-free conditions and the health status of mouse lines were routinely checked by veterinary staff. Animals were sacrificed at the indicated time by CO₂ asphyxia, and blood samples and tissues were collected. All performances on animals were approved by the Animal Care and Use Committee of Zhejiang University as per the Chinese Guidelines for The Care and Use of Laboratory Animals.

**Preparation and characterisation of IOX1 encapsulated liposomes (IOXL) and the mixed liposomes IPLD.** The fabrication procedure of IOXL is as follows: 11.89 mg DOPE, 2.11 mg cholesterol, and 5.69 mg DSPE-mPEG2000 were dissolved in 12 ml chloroform and 4.5 ml methanol. The solution was dried by rotary evaporation to obtain a thin lipid film. IOX1 (5 mg) dissolved in 10 wt% Na₂CO₃ solution (5 ml) was added to the lipid film, stirred overnight at room temperature followed by dialysis against the phosphate buffer solution (PBS, 10 mM) to remove the free IOX1. The obtained liposome, named IOXL, had an IOX1-loading content of 20.1 wt%.

The mixed liposome solution of IOXL with commercial PLD (LIBOD®, DOX loading content 11.1 wt%) was prepared by mixing IOXL and PLD at an IOX1 to DOX molar ratio of 5: 1. The resulting mixed liposome, named IPLD, had an IOX1 concentration of 0.75 mg ml⁻¹ and DOX concentration of 0.5 mg ml⁻¹. The sizes and zeta potentials of the liposomes were measured by Zetasizer Nano-ZS (Malvern Instruments, UK). The data were collected using Zerasizer software 7.02. Their morphologies were observed by a transmission electron microscope (TEM, JEM-1200EX, Japan).

**In vitro cellular uptake.** The cells were cultured in 6-well plates at a density of 5 × 10⁵ cells per well in 2 ml medium overnight and then treated with DOX, DOX + IOX1, PLD, or IPLD for 4 or 8 h. The fluorescence images of the cells were taken using a confocal laser scanning microscope (CLSM, Nikon A1, Japan). The data were collected and analyzed by NIS-Elements AR 3.2 software. The cells were collected after 3 min digestion and analyzed using flow cytometry (BD FACSCalibur™, USA). BD CellQuest™ Pro Software (BD Biosciences) was used for data collection and FlowJo V4.5 was used for data analysis.

**In vitro cytotoxicity.** The cytotoxicity of IOX1, DOX, their combination, or their liposomes was determined by a 3-(4, 5-dimethylthiazol-2-yl)-2, 5-diphenyltetrazolium bromide (MTT) cell proliferation assay based on a modified manufacturer's protocol. Briefly, cells were cultured in 96-well plates at a density of 5 × 10³ cells per well and incubated for 24 h, followed by the addition of different treatments at the set concentrations. After incubation for 48 h, MTT reagent (5 mg ml⁻¹, 20 μl) was added into each well, and the plates were incubated at 37 °C for 4 h. Formazan crystals in each well were dissolved in 100 μl DMSO, and the absorbance at 562 nm was measured using a microplate spectrophotometer (SpectraMax M2E, Molecular Device, Inc., USA). The data were collected by softMax Pro V5.4. Each sample concentration was performed in triplicate, and three independent experiments were performed. The MTT assays of IOX1 combined with other chemotherapeutic drugs (DNR, EPI, or OXA) were performed similarly.

**ATPase assay.** ATPase activity was measured with a Pgp-Glo™ Assay System (Promega, Madison, USA). Briefly, the membrane fractions in assay buffer containing MgATP (5 mM) were added in a 96-well plate. Different concentrations of IOX1 from the stock solutions and verapamil (50 μM) were added. Untreated, Na₃VO₄-treated control and verapamil-treated positive control were also included. After 40 min incubation at 37 °C, the plate was placed on ice to stop the reaction and ATP detection reagent (50 μl) was added to measure the left ATP. The luminescence of each well was determined with a luminometer (FB 12 Luminometer, Titertek Berthold, Germany).

**Immunofluorescence.** Cells were seeded on a Petri dish at a density of 2 × 10⁴ cells per well overnight and then cultured with different treatments at the set concentrations. After timed intervals, the cells were fixed with 4% paraformaldehyde for 20 min and permeabilized in 0.1% Triton X-100 for 10 min. After washing with PBS, the cells were incubated with anti-LC3B (1: 200), anti-CRT (1: 100), or anti-HMGB1 (1: 100) antibody at 4 °C overnight, followed by Alexa Fluor 488- or Alexa Fluor 647-conjugated secondary antibody (1: 200) at 37 °C for 30 min. The nuclei were stained with DAPI. The cells were observed with a CLSM.

For tumour tissue immunofluorescence, the tumour was resected from the treated BALB/c mice and embedded into the OCT block and frozen for cryostat sections. The obtained sections (8 μm thick) were fixed with 4% paraformaldehyde for 15 min. After washing with PBS, they were incubated with the blocking solution (3% BSA, 0.5% Triton X-100 in PBS) for 30 min and stained with the primary antibodies against CRT (1: 100), PD-L1 (1: 100), CD8 (1: 50), or GB (1: 50) in the antibody reaction buffer at 4 °C overnight, followed by incubation with Alexa Fluor 488- or Alexa Fluor 647-conjugated secondary antibody (1: 200) at room temperature for 60 min. DAPI was used for nucleus staining. TUNEL assay was performed on the obtained cryostat sections using a one step TUNEL apoptosis assay kit, following the manufacturer's protocol.

**Flow cytometry for PD-L1 and CRT analysis.** Briefly, cells were cultured in 6-well plates at a density of $3 \times 10^5$ cells per well overnight and then incubated with the treatments at the set concentrations for a timed interval. The cells were collected and subjected to primary antibodies staining (PD-L1, 1: 100; CRT, 1: 100). For detection of total proteins, cells were fixed in 4% paraformaldehyde for 10 min and permeabilized with 0.1% Triton X-100 for 20 min before primary antibody staining. After 30 min incubation, the cells were washed with PBS and incubated with Alexa Fluor 647-conjugated secondary antibody (1:200) for another 30 min. After washing, the cells were analyzed using flow cytometry to determine the membrane or total PD-L1 expression. For CRT detection, the cells were stained with propidium iodide (PI) before analysis and the fluorescence intensity of the stained cells was gated on PI-negative cells.

**ATP detection.** Extracellular secretion of ATP was studied using the ATP detection kit. In brief, cells were seeded in the 24-well plates ($2 \times 10^4$ cells/well) for 12 h, and then incubated with the treatments at the set concentrations. After 4 h incubation, the cell culture supernatant was collected, and the ATP content was tested using the ATP detection kit according to the manufacturer's instructions.

**In vitro DC maturation.** Bone marrow-derived dendritic cells (BMDCs) were obtained from the bone marrow of 6-week-old female BALB/c mice and cultured in 6-well plates in fresh 1640 medium (2 ml) with GM-CSF (20 ng ml⁻¹) and IL-4 (10 ng ml⁻¹). The maturation of BMDCs was studied using a Transwell system. Typically, CT26 cells ($1 \times 10^5$ cells/well) and BMDCs ($5 \times 10^5$ cells/well) were seeded in the transwells and 12-well culture plates, respectively. After 24 h, CT26 cells were first cultured with different treatments for 1 day and then co-incubated with BMDCs for 24 h. Afterwards, the BMDCs were collected for antibodies staining, and the mDCs (CD11c⁺CD80⁺CD86⁺) were sorted on flow cytometry. The pro-inflammatory cytokines including TNF-α and IL-6 in the BMDCs' supernatants were measured by ELISA kits with a standard protocol.

The direct effect of IOX1 on DC maturation was studied by treating BMDCs ($5 \times 10^5$ cells/well) with IOX1 (25 μM) for 24 h. BMDCs and their supernatants were collected respectively for analysis using flow cytometry or ELISA kits.

**Western blotting analysis.** For western blotting analysis, RIPA lysis buffer was used for total protein extraction. Cytoplasmic proteins and nuclear proteins were separated by EpiQuik™ Nuclear Extraction Kit (Epigentek, Farmingdale, USA). The membrane proteins were isolated by membrane and cytosol protein extraction kit. The protein concentration of each sample was measured by BCA kits. After SDS-PAGE, the proteins in the cell lysates were separated and then transferred to a PVDF membrane. The membrane was blocked with 5% non-fat powdered milk in TBST buffer for 1 h and incubated with diluted primary antibodies to JMJD1A (1:1000), β-Catenin (1:4000), PD-L1 (1: 1000), P-gp (1: 1000), p-PERK (1: 1000), eIF2α (1: 1000), p-eIF2α (1: 1000), CRT (1: 1000), pan-Cadherin (1: 10000), β-Actin (1: 10000) or GAPDH (1: 10000) at 4 °C overnight. After extensive washing with TBST buffer, the membrane was incubated with a horseradish peroxidase-coupled secondary antibody (goat anti-rabbit or goat anti-mouse) at room temperature for 1 h. After washing away the secondary antibodies, the membrane was visualised with a chemiluminescence imaging system (CLiNX Science instruments, China). The grey value ratio of the target band to the internal reference analyzed by ImageJ 1.52a represents the relative protein expression.

**Quantitative PCR.** For qPCR assays, the total RNA was extracted from the collected cell lysates using E.Z.N.A® HP Total RNA Kit (OMEGA Bio-Teck, Norcross, USA). cDNA was synthesised from purified RNA using PrimeScript™ RT reagent Kit (Takara, Kusatsu, Japan), and qPCR was performed on a real-time PCR machine (Applied Biosystems StepOne™, USA). The relative quantitation values were calculated using the $2^{-\Delta\Delta CT}$ method and presented as fold-changes. The expression of the target gene was normalised to that of *Gapdh* mRNA. The primers sequences are listed in Supplementary Table 1.

**In vitro T cell proliferation.** The mixed leukocyte reactions were carried out to study the T cell proliferation in vitro. In short, CT26 cells were cultured in 6-well plates ($2 \times 10^5$ cells/well) overnight and then incubated with the treatments at the set concentrations for 24 h. Peripheral blood mononuclear cells (PBMCs) were obtained from the blood of BALB/c mice by density gradient centrifugation. After

CFSE staining, PBMCs were added to the pre-treated CT26 cells at a density of $4 \times 10^5$ cells per well. Three days later, PBMCs in the medium were collected and analyzed by flow cytometry.

For evaluation of IOX1's direct effect on T cell proliferation, PBMCs labeled with CFSE were cultured in 6-well plates ($4 \times 10^5$ cells/well) followed by IOX1 (25 μM) treatment for 3 days. Afterwards, PBMCs were collected for flow cytometry analysis.

**Apoptosis analysis with/without PBMCs.** CT26 cells were cultured in 6-well plates at a density of $2 \times 10^5$ cells per well overnight and then incubated with the treatments at the set concentrations for 24 h. The medium was then replaced with fresh medium, followed by the addition of PBMCs ($4 \times 10^5$ cells/well). After 3 days co-culture, the CT26 cells were collected by trypsinization, resuspended in 500 μl of PBS, and then stained with PI (10 μl) and Annexin V-FITC (5 μl) for 15 min. Flow cytometry was used to analyse the apoptosis rate of each sample. Cell apoptosis without PBMCs co-culturing was also carried out for comparison.

**Pharmacokinetic profiles and biodistribution of IOXL and IPLD.** Six-week-old female BALB/c mice were i.v. injected with IOX1, IOXL, PLD, or IPLD at an equivalent dose of 7.5 mg kg⁻¹ IOX1, and 5 mg kg⁻¹ DOX. At the timed intervals, the blood samples were collected via the orbital venous plexus of the mice. The blood samples (100 μl) were mixed with an equal volume of methanol overnight to ensure the dissolution of IOX1. Then the mixtures were centrifuged ($2500 \times g$, 5 min) to obtain the supernatants. The concentrations of DOX and IOX1 in the supernatants were determined by high performance liquid chromatography (HPLC, Agilent 1260, USA) according to the standard curve, and the blood concentrations were calculated accordingly.

For biodistribution study, BALB/c mice bearing s.c. 80 mm³ CT26 tumours (female, 6 weeks) were i.v. injected with the formula, as described above. After 24 h, the mice were sacrificed, and the major organs (tumours, hearts, livers, spleens, lungs, and kidneys) were harvested and weighed. Subsequently, the obtained tissues were homogenized in a double volume of a solution, which contained PBS and methanol at a volume ratio of 1: 1. The mixtures were ultrasonicated, stood at 37 °C overnight and centrifuged ($2500 \times g$, 5 min) to obtain the supernatants. The concentrations of DOX and IOX1 were determined using HPLC.

**Antitumour efficacy against s.c. CT26 tumours in BALB/c or nude mouse models.** Six-week-old female BALB/c or nude mice were inoculated with CT26 cells ($5 \times 10^5$ cells per mouse) via s.c. injection into the left flank. When the tumours reached about 80 mm³, the mice were randomly grouped ($n = 8$) and i.v. injected with (1) PBS, (2) IOXL, (3) PLD, or (4) IPLD at an equivalent dose of 7.5 mg kg⁻¹ IOX1 and 5 mg kg⁻¹ DOX on every 2 days for 3 times. The length ($L$) and width ($W$) of the tumours, as well as the body weights of the mice, were recorded. The tumour volume ($V$) was calculated using the formula $V$ (mm³) $= L$ (mm) $\times W^2$ (mm²) $\times 0.5$. The mice were euthanized once their tumours exceeded 2500 mm³ per the regulations of the Animal Care and Use Committee of Zhejiang University.

For the large-tumour experiment, 6-week-old female BALB/c mice were s.c. inoculated with CT26 cells ($5 \times 10^5$ cells per mouse) in the left flank. After 10 days, the tumour volume reached about 350 mm³, and the mice were then randomly grouped ($n = 8$). The following steps were the same as described above.

**Blood biochemical assay.** BALB/c mice were i.v. injected with PBS or IPLD at an equivalent dose of 7.5 mg kg⁻¹ IOX1 and 5 mg kg⁻¹ DOX on every 2 days for 4 times. The blood was collected on the day after the last treatment for analysis. The blood of mice in the recovery group was also collected and tested on the 7 days post-recovery.

**In vivo antitumour immune response.** Six-week-old female BALB/c mice with s.c. CT26 tumours of about 80 mm³ were randomly grouped ($n = 3$) and i.v. injected with (1) PBS, (2) IOXL, (3) PLD, or (4) IPLD at an equivalent dose of 7.5 mg kg⁻¹ IOX1 and 5 mg kg⁻¹ DOX on every 2 days for 3 times. Seven days after the last treatment, the TDLNs were harvested, ground, and then centrifuged to obtain a cell suspension. DCs were stained with FITC anti-CD11c (1:250), APC anti-CD86 (1:100), PE anti-CD80 (1:50) antibodies, and then analyzed using flow cytometry.

The harvested tumours were dissected into small pieces and immersed in the solution of 1 mg ml⁻¹ collagenase IV and 0.2 mg ml⁻¹ DNase I at 37 °C for 1 h. The mixtures were filtered to obtain single-cell suspension solutions. The cells were washed with PBS and stained with APC anti-CD3 (1:400), PE anti-CD4 (1:200), FITC anti-CD8 (1:50), Alexa Fluor 647 anti-FOXP3 (1:100), FITC anti-CD45 (1:200), APC anti-F4/80 (1:100) or PE-Cy7 anti-CD206 (1:100) antibodies according to the manufacturer's protocols. The regulatory T cells (Tregs, CD4⁺FOXP3⁺), cytotoxic T lymphocytes (CTLs, CD3⁺CD4⁻CD8⁺), and helper T cells (Ths, CD3⁺CD4⁺CD8⁻), tumour-associated M2 macrophages (CD45⁺F4/80⁺CD206⁺) were gated on the flow cytometry. The positive rate of CTLA-4 on Tregs was measured by PE-Cy7 anti-CD152 (1:100) antibody staining. The spleens were also harvested and stained following the protocols to analyse Tregs, CTLs, and Ths. In addition, lymphocytes in spleens were stained with FITC anti-CD4

(1:200), PerCP/Cy5.5 anti-CD44 (1:100), and APC anti-CD122 (1:100) antibodies to analyse central memory T cells ($T_{CM}$, CD4$^+$CD44$^+$CD122$^+$) by flow cytometry.

The blood was collected from the treated mice on 7 days post-injection and used to analyse the systemic cytokines, including interferon-gamma (IFN-γ), tumour necrosis factor-alpha (TNF-α), interleukin-2 (IL-2), interleukin-6 (IL-6), and interleukin-10 (IL-10) using the corresponding ELISA kits according to the manufacturer's protocols.

**Tumour rechallenge.** CT26 tumour-bearing female BALB/c mice (about 80 mm$^3$) were randomly grouped ($n = 5$) and treated as mentioned above. Three days after the last treatment, the tumours of the mice except for the IPLD-treated group were carefully dissected, and the wounds were sutured. IPLD-treated mice needed no such surgery as the tumours disappeared gradually. On day 30, CT26 cells ($5 \times 10^5$) were re-injected into the left flank, or $^{Luci}$CT26 cells were injected via the tail vein. The volumes of re-implanted s.c. tumours were recorded every 2 days. The lung tumours were monitored by the in vivo imaging system (IVIS Lumina XRMS Series III, PerkinElmer, USA) on every 6 days and the data were collected and analyzed by Living Image® 4.5.2 software.

**Antitumour activity in 4T1 orthotopic and lung metastatic dual tumour models.** The orthotopic 4T1 tumour model was established by s.c. injecting $1 \times 10^6$ 4T1 cells into the left mammary gland of 6-week-female BALB/c mice. When the tumour volume reached about 100 mm$^3$ the mice were randomly grouped ($n = 5$) and treated with PBS, IOXL, PLD, or IPLD at an equivalent dose of 7.5 mg kg$^{-1}$ IOX1 and 5 mg kg$^{-1}$ DOX on every 2 days for 3 times. Three days after the last treatment, $5 \times 10^5$ $^{Luci}$4T1 tumour cells were injected via the tail vein into the mice. On day 22, the lung metastasis tumours were imaged by the IVIS. The mice were sacrificed, and their lungs were dissected and immersed in a fixative solution. The metastatic sites in the lungs were counted through microscopic observation.

**Statistical analysis.** Statistical analysis was assessed using GraphPad Prism (7.0) and Microsoft Excel 2016. Two-tailed Student's $t$ test was used to calculate the statistical significance of two populations, and the log-rank (Mantel–Cox) test was used to compare the survival curves. The specifics of the statistical tests and number of replicates were stated in the figure legends. Results are expressed as means ± SD. The threshold for significance in all tests was $P < 0.05$.

**Reporting summary.** Further information on research design is available in the Nature Research Reporting Summary linked to this article.

## Data availability

The authors declare that the main data supporting the findings of this study are available within the Article and its Supplementary Information or from the authors upon reasonable request. Source data are available as a Source Data file. Source data are provided with this paper.

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

## Acknowledgements

We thank the National Natural Science Foundation of China (51833008 and 21704090), the Zhejiang Key Research Program (2020C01123), the National Key Research and Development Program (2016YFA0200301) for financial supports.

## Author contributions

Y.Q.S. conceived and supervised the research. J.L. and Z.H.Z. conduct the experiments. J.L., Z.H.Z., N.S.Q., Q.Z., G.W.W. and Y.Q.S. interpreted the data and wrote the paper. H.P.J. and Y.P. provided guidance for biological experiments. Z.X.Z. and J.B.T. helped prepare the paper. All authors contributed to results discussion and revision of paper.

## Competing interests

The authors declare no competing interests.

## Additional information

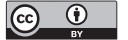

