## [Peer Review File · Nature Communications]

REVIEWER COMMENTS

Reviewer #1 (Remarks to the Author): with expertise in ICD/autophagy/immunology

Potentiating Immunogenic Cell Death (ICD) is central aim of cancer immunotherapy. Jing Liu et al show that a combination of IOX1 (a demethylase and 2-OG oxygenase inhibitor) and Doxorubicin (DOX) treatment (with/without encapsulation in liposomes) resulted in increased apoptosis of cancer cells in vitro and better tumor clearance in vivo in the rather immunogenic CT26 and less immunogenic 4T1 tumor models. Overall, IOX1 improved ICD potential of DOX by reducing gene and protein expression of PD-L1 (a bona-fide checkpoint inhibitor) on cancer cells. A decreased expression of PD-L1 was accompanied by increased DC maturation, T cell infiltration and increased expression of granzyme B from tumor infiltrating T cells. Authors also report a reduction in number of tumor resident regulatory T cells. A combination of IOX1 and DOX not only cleared existing tumors, but also prevented growth of tumors upon re-introduction of CT26 cancer cells in mice. Overall, the findings are interesting and worth but the authors should take more into consideration the mechanistic insights explaining the improved antitumor effects of the combination IOX1+DOX. In the absence of more mechanistic details, some of the conclusions the authors draw seem overstatements.

Main Comments:

The study unfortunately lacks some mechanistic insights at different levels.

What is the actual concentration of IOX1 once it is incorporated into the liposomes?

What is the molecular mechanism of the increased cytotoxicity of IOX1+DOX? Is it simply by increasing DOX concentration in the cells by blockade of the P-gp? The author claims that this correlates with increased LC3 granularity but what is the connection with autophagy? Is this an attempt to resist to excessive cellular stress or is mechanistically involved in cytotoxicity? The conclusion that "IOX1 or its liposome formulation induces Immunogenic apoptosis due to the promoted autophagy" is not supported by the data the authors show. Related to this; the increase in LC3B area has been directly linked to activation of autophagy, but it could merely be the result of an inhibition of the lysosomal degradation of autophagosomes.

Immunofluorescence images of Fig. 1 show that upon DOX+IOX1 and IDOXIL the intracellular level of CRT is increased, not per se its surface redistribution, as shown in Fig 1J. Increased levels of CRT could be a sign of the induction of ER stress and activation of the UPR. Which is a hallmark of ICD. It would be important to check whether the combo DOX+IOX1 increases ICD hallmark because of exacerbated ER stress (particularly PERK pathway).

Figure 2: Was effect of IOX1 on isolated PBMCs tested? In figure 2i, although authors show that apoptosis in CT26 cells was increased upon co-incubation with PBMCs, it would be important to know if proliferation in T cells (in figure 2g) was driven by signals originating from CT26 cells or IOX1.

Figure 2D: Can the author please explain why IDOXIL treatment in CT26 on western blotting shows a decrease of PD-L1 on total cell lysate but an increase on flow cytometry?

To conclude that DAMPs are important for DC maturation, authors should assess the effects of the neutralization of DAMPs in their co-incubation assays, e.g. CRT by neutralizing antibody ecc.

Figure 4. This reviewer finds it hard to believe that in PBS or IOXIL stained tumors authors cannot detect CRT. CRT is a housekeeping protein, so it is constitutively expressed by all cells. It has also been quite hard in the field of ICD, to distinguish the surface expression (bona fide marker of ICD) from the intracellular ER luminal localization of CRT exactly because of the overall staining of CRT in cancer cells (and all cells). Can the authors explain the lack of CRT immunoreactivity in untreated tumors?

Authors show accumulation of IOX1L and IDOXIL in the cancer cells in vitro, but not in vivo, it is assumed that increased TUNEL is due to increased cancer cell death, but it could be the results of CTL activity.

In vivo, authors measure some very interesting effects, including increased cell death (TUNEL) and block of tumor regrowth by the IDOXIL, correlating with increased infiltration of CTL and Grz

staining and the induction of immunological memory, but actually the authors do not show whether this is an effect caused by directly forcing ICD (see comment about CRT staining) in cancer cells. For example, can the author exclude direct effects on immune cells? e.g. TAMs or T cells? Can IOXIL or IDOXIL alter directly the metabolic-epigenetic axis in T cells thus changing their activity status?

Finally, in many figures it is not very clear whether N=3 means independent biological repeats? The very small SD reported, would rather suggest that they are 3 technical repeats. Also pls indicate which statistical test has been applied. One-way Anova and not T test should be applied for multiple comparisons.

Minor

Figure 1f: please show a data bar for IOX1 only instead of combination with DOX regarding P-gp expression.

2e: GAPDH western blot seems overexposed.

Were gene or protein expression of another checkpoint inhibitor like CTLA4 quantified? This would be important to conclude if anti-tumor activity was due to PD-L1 inhibition only.

Supplementary figure 11 – images seem to be overexposed

It is mentioned that small molecules could be more cost-effective in comparison to antibody-based therapy. What evidence do you have to support the potential cost-effectiveness superiority of small molecule inhibitors over antibodies? The mentioned references mention the indeed high cost of antibodies (24) and biological efficacy of some small molecule compounds, not the economical efficacy (25). For example the small molecule inhibitor palbociclib, discussed here: doi: 10.1093/annonc/mdx201, shows 0% chance of being cost effective when being added to breast cancer therapy, with a high cost per QALY gained.

Regarding the ICB induced type I diabetes, this is a relatively rare complication affecting 0,9% of patients treated with ICB ([https://doi.org/10.1016/S2213-8587\(19\)30072-5](https://doi.org/10.1016/S2213-8587(19)30072-5)) However useful to report pancreatic PD-L1 (maybe in regard to upcoming pancreatic cancer treatment?), lung PD-L1 might be just as useful from the sole regard of systemic toxicity since pneumonitis (and others) are slightly more frequently side effects of ICB's in general (<https://doi.org/10.1002/cncr.31043>).

Reviewer #2 (Remarks to the Author): with expertise in IOX1

Manuscript by Liu et al describes the potential use of small molecule IOX1 in cancer immunotherapy. The authors show IOX1 to effectively and selectively downregulate tumour PD-L1 expression in vitro and in vivo. IOX1 inhibited cancer multidrug resistance and enhanced DOX-induced ICD which remodelled the tumour-immunosuppressive microenvironment including increased DC maturation and T-cell infiltration and activity, and reduced Treg cells. IOX1/DOX liposomes eradicated murine tumours, including subcutaneous, orthotopic and lung metastasis tumours, and offered long-term immunological memory function against subcutaneous and lung metastasis re-challenging.

Overall the manuscript is well written and the studies well designed. While the results are somewhat surprising given the promiscuous nature of IOX1 (inhibits a wide-range of 2OG oxygenases and likely other targets), solid data support their observations and important conclusions. IOX1 is a small molecule (better tumour penetration) and in the form of IOXIL appears to be a low-cost and safe alternative to anti-PD1/PD-L1 cancer immunotherapy in combination with DOX, thus presents a novel and exciting finding. However, the mechanism and application of IOX1-mediated increased intracellular DOX levels and reduced PD-L1 mRNA levels in cancer requires more characterisation; understanding the mechanism of action would be essential in exploiting the findings of this study.

- What is the mechanism of IOX1-mediated increased intracellular levels of DOX in CT26 cells?
- Enhanced DOX cytotoxicity at cellular IC50 16nM IOX1 suggests a mechanism independent of KDM / 2OG oxygenase inhibition (e.g. most IC50s are high uM to observe significant changes in histone methylation levels / HIF upregulation)
- Does IOX1 inhibit recombinant P-gp ATPase in vitro activity?
- Does IOX1 increase intracellular DOX in mutant P-gp CT26 cells?
- Why are there variations in IOX1-mediated enhanced DOX cytotoxicity between CT26, 4T1 & B16F10 cell lines? – is it associated with cell line specific P-gp expression / activity?
- Why did IOX1 not enhance DOX cytotoxicity in NIH-3T3 and HUVEC non-cancer cell lines?
- Is the P-gp expression levels / activity in NIH-3T3 and HUVEC similar to CT26?
- Does IOX1 increase intracellular levels of other chemotherapeutic drugs?

Minor point

Line 76: 'histone demethylase inhibitor used for the treatment of b-thalassemia³⁰'; This paper reports the potential for use as treatment – IOX1 is not used as a treatment of b-thalassemia

Reviewer #3 (Remarks to the Author): with expertise in liposomes and Doxorubicin (drug delivery)

The manuscript by Liu and colleagues describes the use of a nanoliposomal histone demethylase inhibitor to potentiate the anticancer activity of liposomal doxorubicin in mice. The authors demonstrate that encapsulated IOX1 can increase doxorubicin accumulation and down regulate PD-L1 expression in mouse cancer cells. In vitro and in vivo results demonstrate impressive antitumor activity of the combined treatment of immunocompetent mice with liposomal doxorubicin and liposomal IOX1. Combined treatment stimulates DC maturation and anti-tumor T cell responses.

The manuscript is mostly well and clearly written and experiments are documented with a large amount of supplemental data. Three major points should be addressed and some minor issues need to be fixed before publication.

1. No data is shown for human cancer cells. Since gene regulation may differ between human and mouse cells, it is critical that the authors show at a minimum that IOX1 can increase dox accumulation and decrease PD-L1 expression in several human cell lines. This paper is only interesting if the results can be translated to humans.
2. The authors tend to overstate the mechanism by which antitumor activity is enhanced. For example, no direct evidence is provided showing that IOX1 affects pgp function. Investigation of cells that overexpress or have pgp KO would be more convincing. Along the same lines, although IOX1 can decrease PD-L1 expression on cancer cells, it probably alters the expression of many genes. The results show a correlation, but do not prove that the results are due to PD-L1 down-regulation. Cells with PD-L1 KO would be useful to prove this. More care in claiming mechanism can solve this problem.
3. Most of the in vitro results use 5 uM DOX and 25 uM IOX1. It is unclear if these doses can be achieved in vivo. The authors do provide tumor concentrations but this appears to be free plus encapsulated drugs. Some discussion or additional data addressing the dose effects of the observations might be useful.

Minor points

1. Probably should add "in mice" to the title to make it clear the results are limited to mice.
2. The first sentences of results is misleading. "IOX1 is used for the treatment of b-thalassemia"

makes it sound like this is a clinical drug. I believe the results for IOX1 are all in experimental models so far.

3. What anti-PD-L1 antibody was used? Was it specific for mouse PD-L1?
4. The authors do NOT use DOXIL in their study. LIBOd is a generic form of PEGylated liposomal doxorubicin which has a different lipid composition and mean size in comparison to DOXIL. All mentions of DOXIL should be replaced with PLD or LIBOd®.
5. The DSL data show that IOX1L ranges from ~ 50 nm to 200 nm. How can it be 102.3 +/- 0.7 nm?
6. There is a lot of data showing particle size stability of the liposomes. This is relatively non-informative. The authors should show drug stability inside the liposomes.
7. The panels are placed in non-intuitive locations in some of the figures. For example, panel f in Figure 1. A more natural flow of panels from right to left and top to bottom is suggested.
8. The effect of IOX1 on rhodamine 123 accumulation in cells appears to be much more dramatic than the effect on DOX. What does that say about the proposed mechanism of action for DOX?
9. The numbers in Figure 1G do not match supplemental fig 10. For example, the IC50 for dox is ~ 1 ug/ml or 1000 ng/ml in the supplemental Fig 10 but is reported as 35 ng/ml in fig 1g for CT26. All values seem to be WAY off.
10. There are too many significant digits for the IC50 values in 1g. Also, no SD values.
11. In the in vitro studies, how do the authors know that some IOX1L is not carried over to DC, and that this is what is causing DC maturation markers to change?
12. How do the authors know that IOX1L are not directly acting on TIL in vivo?
13. The cancer cell lines should be checked for mycoplasma contamination.

Point by point response:

We are grateful to receive valuable comments on our manuscript. We appreciated all three experienced reviewers for their positive reception of our work and their constructive suggestions. Based on the reviewers' suggestions, we performed additional experiments and revised our manuscript accordingly. The major changes are highlighted in yellow. We hope the revision is clear enough to address the major point raised by the reviewers. As detailed below, we addressed each specific point raised by the reviewers.

Please note: As required by Reviewer # 3, we replaced DOXIL with PLD (pegylated liposomal doxorubicin), and IDOXIL with IPLD.

Reviewer # 1

Overall comments:Overall, the findings are interesting and worth but the authors should take more into consideration the mechanistic insights explaining the improved antitumour effects of the combination IOX1+DOX. In the absence of more mechanistic details, some of the conclusions the authors draw seem overstatements.

Response: Thank you for your recognition of our work and valuable suggestions. We elucidated the mechanisms, as required. Please find our itemized responses below.

Q1) What is the actual concentration of IOX1 once it is incorporated into the liposomes?

Response: The IOX1 concentrations in IOXL and IPLD were 1 mg ml⁻¹ and 0.75 mg ml⁻¹ respectively. The information is added in the supplementary materials.

Q2) What is the molecular mechanism of the increased cytotoxicity of IOX1+DOX? Is it simply by increasing DOX concentration in the cells by blockade of the P-gp? The author claims that this correlates with increased LC3 granularity but what is the connection with autophagy? Is this an attempt to resist to excessive cellular stress or is mechanistically involved in cytotoxicity? The conclusion that "IOX1 or its liposome formulation induces Immunogenic apoptosis due to the promoted autophagy" is not supported by the data the authors show. Related to this; the increase in LC3B area has been directly linked to activation of autophagy, but it could merely be the result of an inhibition of the lysosomal degradation of autophagosomes.

Response: We thank the reviewer's important instruction.

- For P-gp: In this revision, we conducted experiments and elucidated the mechanism: IOX1 downregulated P-gp expression in a dose-dependent manner through JMJD1A/β-catenin/P-gp signaling pathway (Figs. 1c,3, and Supplementary Fig. 7). IOX1 also directly inhibited the P-gp function as demonstrated in P-gp ATPase inhibition study (Supplementary Fig. 8). Thus, the increased cytotoxicity is due to the elevated intracellular DOX accumulation resulting from the P-gp downregulation and function inhibition.
- For autophagy: DOX can induce autophagy in cancer cells¹, resulting in apparent LC3 signals (Supplementary Fig. 19). On the contrary, IOX1 treatment did not change the LC3 levels in CT26 cells, suggesting IOX1 alone neither increase the autophagic flux nor inhibit the lysosomal degradation of autophagosomes. However, 5IOX1+DOX treatment increased the LC3 granularity and autophagy-associated ATP secretion² (Fig. 1i) in CT26 cells, indicating the increased LC3 area was caused by the IOX1-mediated enhanced intracellular DOX levels, *i.e.*, excessive cellular stress.

Q3) Immunofluorescence images of Fig. 1 show that upon DOX+IOX1 and IPLD the intracellular level of CRT is increased, not per se its surface redistribution, as shown in Fig 1J. Increased levels of CRT could be a sign of the induction of ER stress and activation of the UPR, which is a hallmark of ICD. It would be important to check whether the combo DOX+IOX1 increases ICD hallmark because of exacerbated ER stress (particularly PERK pathway).

Response: We are grateful for your valuable suggestion.

- We isolated membrane protein and found the membrane CRT protein level was much higher after treating with 5IOX1+DOX, confirming the CRT membrane translocation.
- As suggested, we analyzed activation of PERK/eIF2 α pathway and found that 5IOX1+DOX or their liposomes did induce much stronger ER stress than DOX or its liposome alone. The data are added in **Fig. 1j and Supplementary Fig. 22**.

Q4) Figure 2: Was effect of IOX1 on isolated PBMCs tested? In figure 2i, although authors show that apoptosis in CT26 cells was increased upon co-incubation with PBMCs, it would be important to know if proliferation in T cells (in figure 2g) was driven by signals originating from CT26 cells or IOX1.

Response: Thank you for your suggestion.

- As suggested, we tested the effect of IOX1 on isolated PBMCs **(added as Supplementary Fig. 40)**. IOX1 had no direct effect on the proliferation index (PI) of PBMCs, indicating that the enhanced proliferation of PBMCs was driven by CT26 cells (with reduced PD-L1 expression) rather than IOX1 itself.

Q5) Figure 2D: Can the author please explain why IPLD treatment in CT26 on western blotting shows a decrease of PD-L1 on total cell lysate but an increase on flow cytometry?

Response: Thank you for your comment.

- The difference of PD-L1 amounts in western blotting and flow cytometry analysis is caused by the two methods detecting PD-L1 in different locations of cells: western blotting determines the total PD-L1 of the whole cells but flow cytometry only determines the PD-L1 on the cell membrane. To confirm this, we treated cells with Triton X-100 to make the cell membrane permeable enabling anti-PD-L1 antibody to label the intracellular PD-L1 as well. The flow cytometry analysis of such cells now is consistent with the results of western blotting. Please see the results in **Fig. 2e and Supplementary Fig. 35**.

Q6) To conclude that DAMPs are important for DC maturation, authors should assess the effects of the neutralization of DAMPs in their co-incubation assays, e.g. CRT by neutralizing antibody ecc.

Response: Thank you for your advice.

- We used an alternative to answer the reviewer's question according to the reference method³. CT26 cells were deprived of CRT by its shRNA and then treated with IOX1+DOX. These cells hardly caused DC maturation **(Supplementary Figs. 28,29)**, proving that CRT, one of the DAMPs, is necessary to DC maturation.

Q7) Figure 4. This reviewer finds it hard to believe that in PBS or IOXL stained tumours authors cannot detect CRT. CRT is a housekeeping protein, so it is constitutively expressed by all cells. It has also been quite hard in the field of ICD, to distinguish the surface expression (bona fide marker of ICD) from the intracellular ER luminal localization of CRT exactly because of the overall staining of CRT in cancer cells (and all cells). Can the authors explain the lack of CRT immunoreactivity in

untreated tumours?

Response: Thanks for your question.

- The CRT expression in tumours of PBS or IOXL-treated mice could be detected but was very weak compared with PLD and IPLD groups. Therefore, in the images given in Fig. 5a, the signals in the PBS and IOX1 groups are hardly seen. As you can see from the following images (Fig. R), once the immunofluorescence in the two groups is seen clearly, those in the PLD and IPLD groups would be overexposed.

Fig. R. Immunostaining of CRT exposure in the tumours.

Q8) Authors show accumulation of IOXL and IPLD in the cancer cells in vitro, but not in vivo, it is assumed that increased TUNEL is due to increased cancer cell death, but it could be the results of CTL activity.

Response: Thank you for your advice.

- CTLs did exert strong effect on the excellent antitumour efficiency. The increased TUNEL was attributed to the CTLs activity in the tumours as well.

Q9) In vivo, authors measure some very interesting effects, including increased cell death (TUNEL) and block of tumour regrowth by the IPLD, correlating with increased infiltration of CTL and Grz staining and the induction of immunological memory, but actually the authors do not show whether this is an effect caused by directly forcing ICD (see comment about CRT staining) in cancer cells. For example, can the author exclude direct effects on immune cells? e.g. TAMs or T cells? Can IOXL or IPLD alter directly the metabolic-epigenetic axis in T cells thus changing their activity status?

Response: Thank you for your suggestion.

- Following your advice, we measured the percentage of M2 macrophages (Supplementary Fig. 64) in tumours after different treatments and found that both IOXL and IPLD had no influence on the contents of M2 macrophages in the tumours.
- We demonstrated that IOX1 inhibited JMJD1A and thus downregulated its downstream β -catenin and subsequent PD-L1 and P-gp expression. The JMJD1A in T lymphocytes was much lower than that in CT26 cells (Supplementary Fig. 48). Therefore, IOX1 can not affect the metabolic-epigenetic axis of T cells. The experiments also confirmed that IOX1 had no direct effect on the proliferation of T cells (Supplementary Fig. 40).

Q10) Finally, in many figures it is not very clear whether N=3 means independent biological repeats? The very small SD reported, would rather suggest that they are 3 technical repeats. Also pls indicate which statistical test has been applied. One-way Anova and not T test should be applied for multiple comparisons.

Response: Thank you for your advice.

- During the studies, all the cell experiments were repeated at least 3 times independently. For animal experiments, the samples were from at least 3 different mice in every group. We

evaluated the significant difference between two groups, so T-test was used in statistical analysis.

Minor points:

Q11) Figure 1f: please show a data bar for IOX1 only instead of combination with DOX regarding P-gp expression.

Response: Thank you for your suggestion and we added the corresponding result.

- As seen in **Fig. 1c and Supplementary Fig. 7**, IOX1 alone could reduce the P-gp expression in CT26 and HCT116 cells in a concentration-dependent manner.

Q12) 2e: GAPDH western blot seems overexposed.

Response: Thank you for your suggestion. We adjusted the GAPDH images to the proper brightness.

Q13) Were gene or protein expression of another checkpoint inhibitor like CTLA4 quantified? This would be important to conclude if anti-tumour activity was due to PD-L1 inhibition only.

Response: Thank you for your advice.

- We measured the CTLA-4 expression on Tregs (CD3⁺CD4⁺Foxp3⁺) in tumours. IOX1 had no effect on CTLA-4 expression (**Supplementary Fig. 63**).

Q14) Supplementary figure 11 – images seem to be overexposed.

Response: Thanks for your suggestion. We adjusted the images in the revised manuscript.

Q15) It is mentioned that small molecules could be more cost-effective in comparison to antibody-based therapy. What evidence do you have to support the potential cost-effectiveness superiority of small molecule inhibitors over antibodies? The mentioned references mention the indeed high cost of antibodies (24) and biological efficacy of some small molecule compounds, not the economical efficacy (25). For example the small molecule inhibitor palbociclib, discussed here: doi: 10.1093/annonc/mdx201, shows 0% chance of being cost effective when being added to breast cancer therapy, with a high cost per QALY gained.

Response: Thanks for your suggestion. We revised our wording in the revised manuscript.

Q16) Regarding the ICB induced type I diabetes, this is a relatively rare complication affecting 0,9% of patients treated with ICB ([https://doi.org/10.1016/S2213-8587\(19\)30072-5](https://doi.org/10.1016/S2213-8587(19)30072-5)). However useful to report pancreatic PD-L1 (maybe in regard to upcoming pancreatic cancer treatment?), lung PD-L1 might be just as useful from the sole regard of systemic toxicity since pneumonitis (and others) are slightly more frequently side effects of ICB's in general (<https://doi.org/10.1002/cncr.31043>).

Response: We appreciate the reviewer's important instruction.

- The PD-L1 expression in the lungs of the treated mice was analyzed, as suggested. There was no difference among the groups (**Supplementary Fig. 66**).

Reviewer# 2

Overall comments:Overall the manuscript is well written and the studies well designed. While the results are somewhat surprising given the promiscuous nature of IOX1 (inhibits a wide-range of 2OG oxygenases and likely other targets), solid data support their observations and important conclusions. IOX1 is a small molecule (better tumour penetration) and in the form of IOXIL appears to be a low-cost and safe alternative to anti-PD1/PD-L1 cancer immunotherapy in combination

with DOX, thus presents a novel and exciting finding. However, the mechanism and application of IOX1-mediated increased intracellular DOX levels and reduced PD-L1 mRNA levels in cancer requires more characterisation; understanding the mechanism of action would be essential in exploiting the findings of this study.

Response: Thank you for taking the time to review this manuscript. We appreciated all your comments and suggestions. We did additional experiments and elucidated the mechanisms, as required. Please find our itemized responses below.

Q1) What is the mechanism of IOX1-mediated increased intracellular levels of DOX in CT26 cells?

Response: Thank you for your comment.

- We did more experiments and elucidated the mechanism and summarized it in the new **Fig. 3**: IOX1 reduced the JMJD1A level, and thus downregulated the PD-L1 and P-gp through the β -catenin signaling. Besides, IOX1 directly inhibited P-gp ATPase activity. Both mechanisms led to the increased intracellular levels of DOX in CT26. The results were added in **Fig. 1c and Supplementary Figs. 7,8**.

Q2) Enhanced DOX cytotoxicity at cellular IC₅₀ 16nM IOX1 suggests a mechanism independent of KDM / 2OG oxygenase inhibition (e.g. most IC₅₀s are high μ M to observe significant changes in histone methylation levels / HIF upregulation)

Response: Thank you for your careful review.

- We are sorry for our mistake in IC₅₀ values. The correct IC₅₀ of IOX1 to enhance DOX cytotoxicity was 163 nM and we corrected it in the revision. Additionally, we found out the mechanism, that is, IOX1 inhibited JMJD1A (also called KDM3A, IC₅₀ value was reported to be 0.1 μ M⁴) and thus downregulated its downstream β -catenin and subsequent PD-L1 and P-gp expression. Please see the response in the Q1.

Q3) Does IOX1 inhibit recombinant P-gp ATPase in vitro activity?

Response: Thank you for your suggestion.

- Yes. IOX1 could inhibit P-gp ATPase activity in a concentration-dependent manner (**Supplementary Fig. 8**).

Q4) Does IOX1 increase intracellular DOX in mutant P-gp CT26 cells?

Response: Thank you for your comment.

- We compared intracellular DOX in P-gp knockdown (**Supplementary Figs. 13,14**) or overexpressing CT26 cells (**Supplementary Figs. 13,15**). IOX1 failed to increase intracellular DOX in both cells.

Q5) Why are there variations in IOX1-mediated enhanced DOX cytotoxicity between CT26, 4T1 & B16F10 cell lines? – is it associated with cell line specific P-gp expression / activity?

Response: Thank you for your suggestion.

- IOX1 inhibits JMJD1A and downregulates β -catenin as well as PD-L1 and P-gp in tumour cells. The JMJD1A levels in these cell lines vary significantly (**Supplementary Fig. 46**), and therefore IOX1 had different effects on the enhanced DOX cytotoxicity.

Q6) Why did IOX1 not enhance DOX cytotoxicity in NIH-3T3 and HUVEC non-cancer cell lines?

Response: The JMJD1A levels in NIH-3T3 and HUVEC cell lines were extremely low (**Supplementary Figs. 46,47**), and thus IOX1 had no effect on them.

Q7) Is the P-gp expression levels / activity in NIH-3T3 and HUVEC similar to CT26?

Response: As shown in **Supplementary Figs. 46,47**, the P-gp expression levels in NIH-3T3 and HUVEC were lower than those in tumour cell lines.

Q8) Does IOX1 increase intracellular levels of other chemotherapeutic drugs?

Response: Thank you for your suggestion.

- We found that IOX1 also significantly increased the cytotoxicity of anthracycline and platinum drugs (**Supplementary Fig. 18**) due to the increased intracellular drug levels.

Minor points:

Q9) Line 76: 'histone demethylase inhibitor used for the treatment of b-thalassemia³⁰'; This paper reports the potential for use as treatment – IOX1 is not used as a treatment of b-thalassemia.

Response: Thank you for your correction. We revised it accordingly.

Reviewer# 3

Overall comments:The manuscript is mostly well and clearly written and experiments are documented with a large amount of supplemental data. Three major points should be addressed and some minor issues need to be fixed before publication.

Response: We greatly appreciate the Reviewer's recognition of our work and great suggestions. As required, we did additional experiments on human colon cancer cell line HCT116 and human breast cancer cell line MCF-7, and validated the conclusions. Please find our itemized responses below.

Q1) No data is shown for human cancer cells. Since gene regulation may differ between human and mouse cells, it is critical that the authors show at a minimum that IOX1 can increase dox accumulation and decrease PD-L1 expression in several human cell lines. This paper is only interesting if the results can be translated to humans.

Response: Thank you for your suggestion.

- As required, two human cell lines were tested. IOX1 could increase DOX accumulation (**Supplementary Figs. 10,16**) and downregulate PD-L1 expression (**Supplementary Figs. 31-33**) in HCT116 cell and MCF-7 human cell lines.

Q2) The authors tend to overstate the mechanism by which antitumour activity is enhanced. For example, no direct evidence is provided showing that IOX1 affects pgp function. Investigation of cells that overexpress or have pgp KO would be more convincing. Along the same lines, although IOX1 can decrease PD-L1 expression on cancer cells, it probably alters the expression of many genes. The results show a correlation, but do not prove that the results are due to PD-L1 down-regulation. Cells with PD-L1 KO would be useful to prove this. More care in claiming mechanism can solve this problem.

Response: Thank you for your suggestions.

- In the revised manuscript, we did additional experiments and proved that IOX1 inhibited both P-gp expression and pump function. As suggested, we constructed P-gp-knockdown CT26 cells by shRNA. IOX1 had no effects on these cells' DOX accumulation (**Please see the Supplementary Figs. 13,14**).
- We elucidated the IOX1 working pathway: IOX1 inhibited JMJD1A and thus downregulated its downstream β -catenin and subsequent PD-L1 and P-gp expression. As required, we

constructed PD-L1-knockdown CT26 cells. These cells with IOX1 treatment lost the ability to promote T-cell proliferation, confirming that PD-L1 downregulation by IOX1 was vital to T-cell proliferation (Please see the Supplementary Figs. 36,37).

Q3) Most of the in vitro results use 5 uM DOX and 25 uM IOX1. It is unclear if these doses can be achieved in vivo. The authors do provide tumour concentrations but this appears to be free plus encapsulated drugs. Some discussion or additional data addressing the dose effects of the observations might be useful.

Response: Thanks for your suggestion.

- From the results of biodistribution assay (Supplementary Fig. 50), the tumour DOX was 10.42 nM and IOX1 was 51.38 nM; the molar ratio of IOX1 to DOX was 4.93:1, close to the dosage ratio (IOX1: DOX = 5:1). Indeed, the drug in tumour was in free and liposomal forms. Although the drug concentration in tumours were lower than the dose *in vitro*, IPLD did induce enhanced ICD and downregulate the PD-L1 levels *in vivo* (Fig. 5a) indicating that IPLD maintained its function *in vivo*.

Minor points:

Q4) Probably should add “in mice” to the title to make it clear the results are limited to mice.

Response: Thank you for your suggestion.

- As mentioned in the first question, we validated our results on human cell lines.

Q5) The first sentences of results is misleading. “IOX1 is used for the treatment of b-thalassemia” makes it sound like this is a clinical drug. I believe the results for IOX1 are all in experimental models so far.

Response: Thank you for your correction. We revised it accordingly.

Q6) What anti-PD-L1 antibody was used? Was it specific for mouse PD-L1?

Response: Yes. We used the anti-PD-L1 antibody (Abcam, ab213480) specific for mouse PD-L1 on murine cells' experiment (<https://www.abcam.cn/pd-l1-antibody-epr20529-ab213480.html>).

Q7) The authors do NOT use PLD in their study. LIBOd is a generic form of PEGylated liposomal doxorubicin which has a different lipid composition and mean size in comparison to PLD. All mentions of PLD should be replaced with PLD or LIBOd®.

Response: Thank you for your correction. We replaced DOXIL with PLD in the revision.

Q8) The DSL data show that IOXIL ranges from ~ 50 nm to 200 nm. How can it be 102.3 +/- 0.7 nm?

Response: The DLS gave the size range from 50-200 nm. The 102.3 nm was the volume-averaged size.

Q9) There is a lot of data showing particle size stability of the liposomes. This is relatively non-informative. The authors should show drug stability inside the liposomes.

Response: Thank you for your suggestion.

- Following your suggestion, we analyzed IOX1 inside the liposomes every three days for 30 days. The drug loading content and the molecular structure did not change (Supplementary Fig. 6c).

Q10) The panels are placed in non-intuitive locations in some of the figures. For example, panel f

in Figure 1. A more natural flow of panels from right to left and top to bottom is suggested.

Response: Thank you for your suggestion. We corrected it as suggested.

Q11) *The effect of IOX1 on rhodamine 123 accumulation in cells appears to be much more dramatic than the effect on DOX. What does that say about the proposed mechanism of action for DOX?*

Response: Rhodamine 123 is a P-gp substrate easier affected by P-gp and ATPase activity. The similar results confirmed the P-gp downregulation and function inhibition by IOX1.

Q12) *The numbers in Figure 1G do not match supplemental fig 10. For example, the IC50 for dox is ~ 1 ug/ml or 1000 ng/ml in the supplemental Fig 10 but is reported as 35 ng/ml in fig 1g for CT26. All values seem to be WAY off.*

Response: Thank you very much for the important correction. We corrected them in the revision.

Q13) *There are too many significant digits for the IC50 values in 1g. Also, no SD values.*

Response: Thank you for your suggestion. We corrected them and added SD values in the revision.

Q14) *In the in vitro studies, how do the authors know that some IOX1L is not carried over to DC, and that this is what is causing DC maturation markers to change?*

Response: Thank you for your suggestions.

- As suggested, we tested the direct effects of IOX1 on DC maturation. As shown in **Supplementary Fig. 27**, IOX1 alone showed no effect on DC maturation as well as its related markers.

Q15) *How do the authors know that IOXL are not directly acting on TIL in vivo?*

Response: According to the flow cytometry results of CTLs (CD3⁺CD8⁺) (Fig. 5f) and IF images of CD8 and granzyme B in tumours (Fig. 5h), IOXL alone showed no contribution to recruitment or activation of CTLs. As IOXL alone showed little antitumour effect on each model, we believe that IOXL has no direct effect on T cells.

Q16) *The cancer cell lines should be checked for mycoplasma contamination.*

Response: Thank you for your suggestion.

- We are sorry for missing the information. Actually, we checked our cell lines for mycoplasma contamination before all experiments. We added the related information in the supplementary materials.

1. Tian, W., Xie, X.J. & Cao, P.L. Magnoflorine improves sensitivity to doxorubicin (DOX) of breast cancer cells via inducing apoptosis and autophagy through AKT/mTOR and p38 signaling pathways. *Biomed. Pharmacother.* **121**, 10 (2020).
2. Michaud, M., *et al.* Autophagy-dependent anticancer immune responses induced by chemotherapeutic agents in mice. *Science* **334**, 1573-1577 (2011).
3. Obeid, M., *et al.* Calreticulin exposure dictates the immunogenicity of cancer cell death. *Nature Medicine* **13**, 54-61 (2007).
4. Schiller, R., *et al.* A cell-permeable ester derivative of the JmJc histone demethylase inhibitor IOX1. *Chemmedchem* **9**, 566-571 (2014).

REVIEWER COMMENTS

Reviewer #1 (Remarks to the Author):

The authors have answered properly several of the reviewers concerns which they have addressed by additional in vitro and in vivo experiments, which together reinforce their conclusions. The in vivo data are convincing.

However, the authors still need to carefully tone down conclusions that are based on correlative data, related to e.g. ICD or their mechanistic analysis in general. Correlation does not mean causality. e.g. the observation that LC3 granularity is increased by the combination of IOX1+DOX does not necessarily prove that autophagy is responsible for ATP release, unless the authors use genetic approach to down regulate autophagy and show that ATP release (what the author shows at 24 h cannot be distinguished from a passive release) is reduced by it. so they should change 'autophagy-dependent ATP release' as increased LC3 granularity was associated to enhanced ATP release".

On the same vein, the authors checked PERK-eIF2a P as parameter of integrated stress response (not as such full blown ER stress, which implicates also IRE1 and ATF6 signaling) and correlate this to increased CRT (by the way, surface CRT by WB should not only show marker of PM but also exclude contamination of cytoplasmic fractions, it remains clear that IOX1+ DOX combo treatment increases the overall endogenous levels of CRT not only its redistribution to the PM). But do not prove that this ISR pathway is responsible for CRT exposure.

Reviewer #3 (Remarks to the Author):

The authors did a good job answering the questions. One concern is the effect seems less when using human cancer cells, but the response is still significant

Reviewer #4 (Remarks to the Author): to replace Reviewer #2.

I think the authors have largely addressed the critiques raised by the previous reviewer #2. But, the proposed mechanism of IOX1 inhibition of JMJD1A for the improved anti-tumor effects of IOX1+DOX combination are unconvincing. (1) IOX1 is a broad-spectrum inhibitor of 2OG oxygenases, including several JmjC demethylase members such as JMJD1A, JMJD3, JMJD2A, JMJD2C, JMJD2E, and others. It is inappropriate to say that IOX1 is JMJD1A inhibitor. (2) IOX1 is a catalytic inhibitor of 2OG oxygenases to inhibit the demethylase activity. It is hard to interpret how IOX1 treatment reduced the JMJD1A protein levels (Figure 3h, 3i). Please discuss how IOX1 reduced the JMJD1A protein levels. (3) The authors showed that knockdown of JMJD1A can inhibits the expression of P-gp and PD-L1 in a beta-catenin dependent manner (Figure 3b to 3g). Unfortunately, the authors did not test whether the JMJD1A-knockdown cells showed similar phenotypes as the IOX1 treatment in vitro and in the mouse models. Thus, it is unclear whether JMJD1A is the IOX1 target for the improved anti-tumor effects of IOX1+DOX combination.

The reviewer will suggest remove the "JMJD1A inhibitor" in the title and other part of manuscript as IOX1 is not selective JMJD1A inhibitors and the authors do not have strong evidence that JMJD1A is the major target of IOX1 for the phenotypes observed in this study. Please lower the tone on JMJD1A and properly discuss the possible IOX1 targets.

Point-by-point responses:

We thank the reviewers for their valuable comments. Please see below point-by-point responses to the comments/queries.

Reviewer # 1

Q1) However, the authors still need to carefully tone down conclusions that are based on correlative data, related to e.g. ICD or their mechanistic analysis in general. Correlation does not mean causality. e.g. the observation that LC3 granularity is increased by the combination of IOX1+DOX does not necessarily prove that autophagy is responsible for ATP release, unless the authors use genetic approach to down regulate autophagy and show that ATP release (what the author shows at 24 h cannot be distinguished from a passive release) is reduced by it. so they should change 'autophagy-dependent ATP release' as increased LC3 granularity was associated to enhanced ATP release".

Response: Thanks for your instruction. We revised the wording as instructed.

Q2) On the same vein, the authors checked PERK-eIF2a P as parameter of integrated stress response (not as such full blown ER stress, which implicates also IRE1 and ATF6 signaling) and correlate this to increased CRT (by the way, surface CRT by WB should not only show marker of PM but also exclude contamination of cytoplasmic fractions, it remains clear that IOX1+ DOX combo treatment increases the overall endogenous levels of CRT not only its redistribution to the PM). But do not prove that this ISR pathway is responsible for CRT exposure.

Response: Thanks for your suggestion. We revised our wording accordingly.

Reviewer # 3

The authors did a good job answering the questions. One concern is the effect seems less when using human cancer cells, but the response is still significant.

Response: We sincerely thank the reviewer for taking the time to review our manuscript.

Reviewer # 4

Q1) IOX1 is a broad-spectrum inhibitor of 2OG oxygenases, including several JmjC demethylase members such as JMJD1A, JMJD3, JMJD2A, JMJD2C, JMJD2E, and others. It is inappropriate to say that IOX1 is JMJD1A inhibitor.

Response: Thanks for your correction. We revised our wording as required.

Q2) IOX1 is a catalytic inhibitor of 2OG oxgenases to inhibit the demethylase activity. It is hard to interpret how IOX1 treatment reduced the JMJD1A protein levels (Figure 3h, 3i). Please discuss how IOX1 reduced the JMJD1A protein levels.

Response: Thanks for your comment. We did additional experiments and found that IOX1 reduced the JMJD1A protein level by promoting JMJD1A degradation without affecting its transcription (Supplementary Fig. 50). The discussion is added in Page 13.

Q3) The authors showed that knockdown of JMJD1A can inhibits the expression of P-gp and PD-L1 in a beta-catenin dependent manner (Figure 3b to 3G). Unfortunately, the authors did not test

whether the JMJD1A-knockdown cells showed similar phenotypes as the IOX1 treatment in vitro and in the mouse models. Thus, it is unclear whether JMJD1A is the IOX1 target for the improved anti-tumor effects of IOX1+DOX combination.

Response: Thanks for your comment.

As required, we did additional *in vitro* and *in vivo* experiments. We constructed JMJD1A-knockdown CT26 and HCT116 cells. *In vitro*, we found that IOX1 could not affect these cells' DOX accumulation (Supplementary Figs. 44,45) and the P-gp and PD-L1 expressions anymore (Supplementary Figs. 46-49). *In vivo*, the xenograft tumours of JMJD1A-knockdown CT26 had reduced protein levels of JMJD1A and its downstream targets, β -catenin, PD-L1 and P-gp (Supplementary Fig. 64), similar to the IOXL-treated ones. Accordingly, PLD alone eradicated these tumours (Supplementary Figs. 62,63). These results confirm that the JMJD1A was the main target of IOX1 for the improved anti-tumour effects of IOX1+DOX combination. The discussion is added in Page 11,12,14.

Q4) The reviewer will suggest removing the "JMJD1A inhibitor" in the title and other part of manuscript as IOX1 is not selective JMJD1A inhibitors and the authors do not have strong evidence that JMJD1A is the major target of IOX1 for the phenotypes observed in this study. Please lower the tone on JMJD1A and properly discuss the possible IOX1 targets.

Response: Thanks for your suggestion. We revised the wording as required.

REVIEWERS' COMMENTS

Reviewer #4 (Remarks to the Author):

The authors have addressed my concerns and the manuscript is ready for publication.